# Variational Elliptical Processes

### Abstract

We present elliptical processes—a family of non-parametric probabilistic models that subsumes the Gaussian processes and the Student's $t$ processes. This generalization includes a range of new heavy-tailed behaviors while retaining computational tractability. The elliptical processes are based on a representation of elliptical distributions as a continuous mixture of Gaussian distributions. We parameterize this mixture distribution as a spline normalizing flow, which we train in two different ways using variational inference. The proposed form of the variational posterior enables a sparse variational elliptical process applicable to large-scale problems. We highlight some advantages compared to a Gaussian process through regression and classification experiments. Elliptical processes can replace Gaussian processes in several settings, including cases where the likelihood is non-Gaussian or when accurate tail modeling is essential.

## 1 Introduction

Systems for autonomous decision-making are increasingly dependent on predictive models. To ensure safety and reliability, it is essential that these models capture uncertainty and risk accurately. Gaussian processes ($\mathcal{GP}$s) offer a powerful framework for probabilistic modeling that is widely used, in part because it provides such uncertainty estimates. However, these estimates are only reliable to the extent that the model is correctly specified, i.e. that the assumptions of Gaussianity hold true. On the contrary, heavy-tailed data arise in many real-world applications, including finance (Mandelbrot, 1963), signal processing (Zoubir et al., 2012) and geostatistics (Diggle et al., 1998). We use a combination of normalizing flows and modern variational inference techniques to extend the modeling capabilities of $\mathcal{GP}$s to the more general class of elliptical processes ($\mathcal{EP}$s).

**Elliptical processes.** The elliptical processes subsume the Gaussian process and the Student's $t$ process (Rasmussen & Williams, 2006; Shah et al., 2014). It is based on the elliptical distribution—a scale-mixture of Gaussian distributions attractive mainly because it can describe heavy-tailed distributions while retaining most of the Gaussian distribution's computational tractability (Fang et al., 1990). We use a normalizing flow (Papamakarios et al., 2021a) to model the continuous scale-mixture, which provides an added flexibility that can benefit a range of applications. We explore the use of elliptical processes as both a prior (over functions) and a likelihood, as well as the combination thereof. We also explore the use of $\mathcal{EP}$s as a variational posterior that can adapt its shape to match complex posterior distributions.

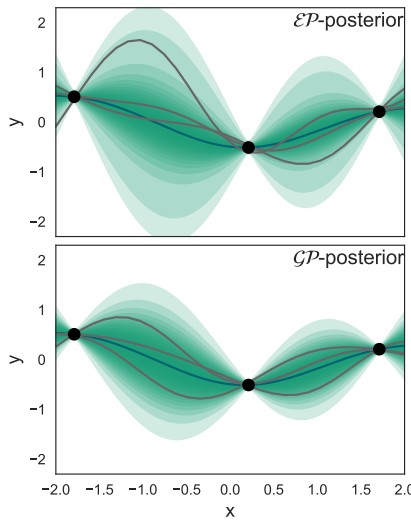

Figure 1: Posterior distributions of an elliptical process and a Gaussian process with equal kernel hyperparameters and covariance. The shaded area are confidence intervals of the posterior processes. The elliptical confidence regions are wider due to the process's heavier tail, which makes the confidence region similar to the Gaussian's close to the mean, but also allows samples further out at the tail.

**Variational inference.** Variational inference is a powerful tool for approximate inference that uses optimization to find a member of a predefined family of distributions that is close to the target distribution (Wainwright et al., 2008; Blei et al., 2017). Significant advances made in the last decade have made variational inference the method of choice for scalable approximate inference in complex parametric models (Ranganath et al., 2014; Hoffman et al., 2013; Kingma & Welling, 2013; Rezende et al., 2014).

It is thus not surprising that the quest for more expressive and scalable variations of Gaussian processes has gone hand-in-hand with the developments in variational inference. For instance, sparse $\mathcal{GP}$s use variational inference to select inducing points to approximate the prior (Titsias, 2009). Inducing points is a common building block in deep probabilistic models such as deep Gaussian processes (Damianou & Lawrence, 2013; Salimbeni et al., 2019) and can also be applied in Bayesian neural networks Maroñas et al. (2021); Ober & Aitchison (2021). Similarly, the combination of inducing points and variational inference enables scalable approximate inference in models with non-Gaussian likelihoods (Hensman et al., 2013a), such as when performing $\mathcal{GP}$ classification (Hensman et al., 2015; Wilson et al., 2016).

However, the closeness of the variational distribution to the target distribution is bounded by the flexibility of the variational distribution. Consequently, the success of deep (neural network) models have inspired various suggestions on flexible yet tractable variational distributions, often based on parameterized transformations of a simple base distribution (Tran et al., 2016). In particular, models using a composition of invertible transformations, known as normalizing flows, have been especially popular (Rezende & Mohamed, 2015; Papamakarios et al., 2021a).

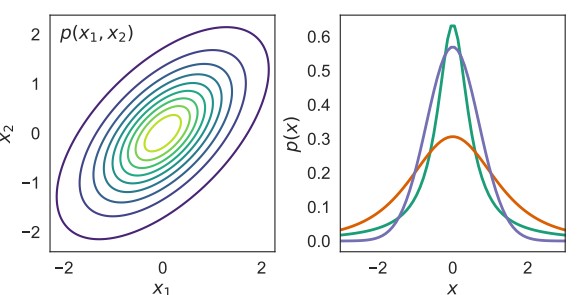

Figure 2: **Left:** A contour plot of an elliptical two-dimensional, correlated distribution with zero means. The name derives from its elliptical level sets. **Right:** Three examples of one-dimensional elliptical distributions with zero means and varying tail-heaviness. Elliptical distributions are symmetric around the mean $\mathbb{E}[\mathbf{X}] = \boldsymbol{\mu}$.

**Our contributions.** We propose an adaptation of elliptical distributions and processes in the same spirit as modern Gaussian processes. Constructing elliptical distributions based on a normalizing flow provides a high degree of flexibility without sacrificing computational tractability. This makes it possible to sidestep the "curse of Gaussianity", and adapt to heavy-tailed behavior when called for. We thus foresee many synergies between $\mathcal{EP}$s and recently developed $\mathcal{GP}$ methods. We make a first exploration of these, and simultaneously demonstrate the versatility of the elliptical process as a model for the prior and/or the likelihood, or as the variational posterior. In more detail, our contributions are:

- a construction of the elliptical process and the elliptical likelihood as a continuous scale-mixture of Gaussian processes parameterized by a normalizing flow;

- a variational approximation that can either learn an elliptical likelihood or handle known non-Gaussian likelihoods, such as in classification problems;

- formulating a sparse variational approximation for large-scale problems, as well as developing and comparing two different training schemes;

- describing extensions to heteroscedastic and multi-path data enabled by amortized variational inference.

## 2 Background

In this section, we present the necessary background on elliptical distributions, elliptical processes and normalizing flow models. Throughout, we consider the regression problem, where we are given a set of $N$ scalar observations, $\boldsymbol{y} = [y_1, \cdots, y_N]^\top$, at the locations $[\boldsymbol{x}_1, \cdots, \boldsymbol{x}_N]^\top$, where $\boldsymbol{x}_n$ is $D$-dimensional. The measurements $y_n$ are assumed to be noisy measurements, such that,

$$y_n = f(\boldsymbol{x}_n) + \epsilon_n, \tag{1}$$

where $\epsilon_n$ is zero mean, i.i.d., noise. The task is to infer the underlying function, $f : \mathbb{R}^D \to \mathbb{R}$.

### 2.1 Elliptical distributions

The elliptical process is based on elliptical distributions (Figure 2), which include Gaussian distributions as well as more heavy-tailed distributions, such as the Student's $t$ distribution and the Cauchy distribution.

The probability density of a random variable $Y \in \mathbb{R}^N$ that follows the elliptical distribution can be expressed as,

$$p(u; \boldsymbol{\eta}) = c_{n,\boldsymbol{\eta}} |\boldsymbol{\Sigma}|^{-1/2} g_N(u; \boldsymbol{\eta}), \tag{2}$$

where $u = (\mathbf{y} - \boldsymbol{\mu})^\mathsf{T} \boldsymbol{\Sigma}^{-1}(\mathbf{y} - \boldsymbol{\mu})$ is the squared Mahalanobis distance, $\boldsymbol{\mu}$ is the location vector, $\boldsymbol{\Sigma}$ is the non-negative definite scale matrix, and $c_{N,\boldsymbol{\eta}}$ is a normalization constant. The density generator $g_{N,\boldsymbol{\eta}}(u)$ is a non-negative function with finite integral parameterized by $\boldsymbol{\eta}$ which determines the shape of the distribution.

Elliptical distributions are consistent, i.e., closed under marginalization, if and only if $p(u; \boldsymbol{\eta})$ is a scale-mixture of Gaussian distributions (Kano, 1994). The density can be expressed as

$$p(u; \boldsymbol{\eta}) = |\boldsymbol{\Sigma}|^{-\frac{1}{2}} \int_0^\infty \left(\frac{1}{2\pi\xi}\right)^{\frac{n}{2}} e^{\frac{-u}{2\xi}} \, p(\xi; \boldsymbol{\eta}_\xi) d\xi, \tag{3}$$

using a mixing variable $\xi \sim p(\xi; \boldsymbol{\eta}_\xi)$. Any mixing distribution $p(\xi; \boldsymbol{\eta}_\xi)$ that is strictly positive can be used to define a consistent elliptical process. In particular, we recover the Gaussian distribution if the mixing distribution is a Dirac delta function and the Student's $t$ distribution if it is a scaled inverse chi-square distribution. For more information on the elliptical distribution, see Appendix A

### 2.2 Elliptical processes

The elliptical process is defined, analogously to a Gaussian process, as:

**Definition 1** *An elliptical process ($\mathcal{EP}$) is a collection of random variables such that every finite subset has a consistent elliptical distribution, where the scale matrix is given by a covariance kernel.*

This means that an $\mathcal{EP}$ is specified by a mean function $\mu(\mathbf{x})$, scale matrix (kernel) $k(\mathbf{x}, \mathbf{x})$ and mixing distribution $p(\xi; \boldsymbol{\eta}_\xi)$. Since the $\mathcal{EP}$ is built upon consistent elliptical distributions it is closed under marginalization. The marginal mean $\boldsymbol{\mu}$ is the same as the mean for the Gaussian distribution, and the covariance is $\mathrm{Cov}[\mathbf{Y}] = \mathbb{E}[\xi] \boldsymbol{\Sigma}$ where $\mathbf{Y}$ is an elliptical random variable, $\boldsymbol{\Sigma}$ is the covariance for a Gaussian distribution and $\xi$ is the mixing variable.

Formally a stochastic process $\{X_t : t \in T\}$ on a probability space $(\Omega, \mathcal{F}, P)$ consists of random maps $X_t : \omega \to S_t$, $t \in T$, for measurable spaces $(S_t, \mathcal{S}_t)$, $t \in T$ (Bhattacharya & Waymire, 2007). We focus on the setting where $S = \mathbb{R}$ and the index set $T$ is a subset of $\mathbb{R}^N$, in particular, the half-line $[0, \infty)$. Due to Kolmogorov's extension theorem, we may construct the $\mathcal{EP}$ from the family of finite-dimensional, consistent, elliptical distributions, which is easy to check due to the restriction to $S = \mathbb{R}$ (which is a Polish space) and Kano's characterization above.

**Identifiability.** When using $\mathcal{GP}$ for regression or classification we usually assume that the data originate from a single sample path, which is a single sample from the $\mathcal{GP}$. An elliptical process, on the other hand, can be viewed as a hierarchical model, constructed by first sampling $\xi \sim p(\xi; \boldsymbol{\eta}_\xi)$ and then $\boldsymbol{f} \sim \mathcal{GP}(\boldsymbol{f}; \boldsymbol{\mu}, \boldsymbol{K}\xi)$. This structure implies that it is not possible to infer the mixing distribution $p(\xi; \boldsymbol{\eta}_\xi)$ from a single path.In other words, the identification condition for the mixing distribution $p(\xi; \boldsymbol{\eta}_\xi)$ is to have draws from multiple paths. This point is explored further in sections 3.4 and 4.5.

**Prediction.** To use the $\mathcal{EP}$ for predictions, we need the conditional mean and covariance of the corresponding elliptical distribution. The conditional distribution is guaranteed to be a consistent elliptical distribution but not necessarily the same as the original one—the shape depends on the training samples. (Recall that consistency only concerns the marginal distribution.) The conditional distribution can be derived analytically (see Appendix B) but we will instead solve it by approximate the posterior $p(\xi | \boldsymbol{y}; \boldsymbol{\eta}_\xi)$ with a variational distribution $q(\xi; \boldsymbol{\varphi}_\xi)$. The approximate inference framework also let us to incorporate (non-Gaussian) noise according to the graphical models in Figure 3.

We aim to model mixing distributions that can capture any shape of the elliptical noise in the data. One way to learn complex probability distributions is to normalize flows, which we will now go through.

### 2.3 Flow based models

*Normalizing flows* are a family of generative models that map simple distributions to complex ones through a series of learned transformations (Papamakarios et al., 2021b). Suppose we have a random variable $\boldsymbol{x}$ that follows an unknown probability distribution $p_x(\boldsymbol{x})$. Then, the main idea of a normalizing flow is to express $\boldsymbol{x}$ as a transformation $T_\gamma$ of a variable $\mathbf{z}$ with a known simple probability distribution $p_z(\mathbf{z})$. The transformation $T_\gamma$ has to be bijective and invertible, and it can have learnable parameters $\gamma$. Both $T$ and its inverse have to be differentiable. The probability density of $\boldsymbol{x}$ is obtained by a change of variables:

$$p_x(\boldsymbol{x}) = p_z(\boldsymbol{z}) \left| \det\left( \frac{\partial T_\gamma(\boldsymbol{z})}{\partial \boldsymbol{z}} \right) \right|^{-1}. \tag{4}$$

We focus on one-dimensional flows, since we are interested in modeling the mixing distribution. In particular, we use *linear rational spline flows* Dolatabadi et al. (2020); Durkan et al. (2019), wherein the mapping $T_\gamma$ is an elementwise, monotonic linear rational spline: a piecewise function where each piece is a linear rational function. The parameters are the number of pieces (bins) and the knot locations.

To train the model parameters, we use amortized variational inference, which we go through next.

### 2.4 Amortized variational inference

In *amortized variational inference* (Gershman & Goodman, 2014) we replace the variational parameters, $\boldsymbol{\varphi}$ with a function that maps the input to the variational parameters $\boldsymbol{\varphi} = g(\boldsymbol{x})$. This is convenient for modelling local latent variables, i.e., variables associated directly to individual data points $\boldsymbol{x}_n$ which have corresponding variational parameters $\boldsymbol{\varphi}_i$. By replacing the local parameter with a function, $\boldsymbol{\varphi}_i = g(\boldsymbol{x}_n)$, we reduce the problem to fitting a function $g$, rather than fitting each $\boldsymbol{\varphi}_i$. Furthermore, it becomes easy to add new data points, since the local variational parameters are then given by the function $g$.

## 3 Method

We propose the variational $\mathcal{EP}$ with elliptical noise, where the variational $\mathcal{EP}$ can learn any consistent elliptical process, and the elliptical noise can capture any consistent elliptical noise. The key idea is to model the mixing distributions with a normalizing flow. The joint probability distribution of the model (see Figure 3c) is

$$p(\boldsymbol{y}, \boldsymbol{f}, \omega, \xi; \boldsymbol{\eta}) = \underbrace{p(\boldsymbol{f}|\xi; \boldsymbol{\eta_f})p(\xi; \boldsymbol{\eta}_\xi)}_{\text{prior}} \underbrace{\prod_{i=1}^{N} p(y_i|f_i, \omega)p(\omega; \boldsymbol{\eta}_\omega)}_{\text{likelihood}}. \tag{5}$$

Here, $p(\boldsymbol{f}|\xi; \boldsymbol{\eta_f}) \sim \mathcal{N}(0, K\xi)$ is a regular $\mathcal{EP}$ prior with the covariance kernel $K$ containing the parameters $\boldsymbol{\eta_f}$, $p(\xi; \boldsymbol{\eta_\xi})$ is the process mixing distribution and $p(\omega; \boldsymbol{\eta_\omega})$ is the noise mixing distribution.

To learn the mixing distributions $p(\xi; \boldsymbol{\eta_\xi})$ and $p(\omega; \boldsymbol{\eta_\omega})$ by gradient-based optimization, they need to be differentiable with respect to the parameters $\boldsymbol{\eta_\xi}$ and $\boldsymbol{\eta_\omega}$ in addition to being flexible and computationally efficient to sample and evaluate. Based on these criteria, a spline flow (Section 2.3) is a natural fit. We construct the mixing distributions by transforming a sample from a standard normal distribution with a spline flow. The output of the spline flow is then projected onto the positive real axis using a differentiable function such as *Softplus* or *Sigmoid*.

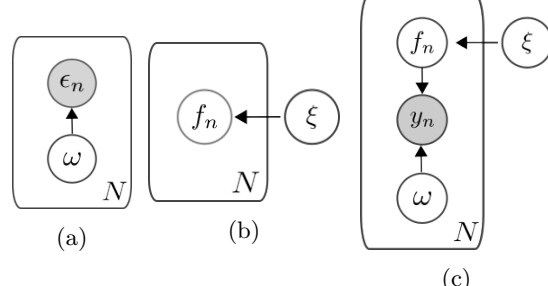

Figure 3: Graphical models of **(a)**, the elliptical likelihood, **(b)** the $\mathcal{EP}$-prior, and **(c)** the $\mathcal{EP}$ with independent elliptical noise.

In the following sections, we detail the construction of the model and show how to train it using variational inference. For clarity, we describe the likelihood first, before combining it with the prior and describing a (computationally efficient) sparse approximation.

### 3.1 Likelihood

By definition, the likelihood (Figure 3a) describes the measurement noise $\epsilon_n$ (Equation (1)). The probability distribution of the independent elliptical likelihood is,

$$p(\epsilon_n; \sigma, \boldsymbol{\eta_\omega}) = \int \mathcal{N}(\epsilon_n; 0, \sigma^2\omega)p(\omega; \boldsymbol{\eta_\omega})d\omega, \tag{6}$$

where $\sigma$ can be set to unity without loss of generality. In other words, the likelihood is a continuous mixture of Gaussian distributions where, e.g., $\epsilon_n$ follows a Student's $t$ distribution if $\omega$ is scaled chi-squared distributed.

**Parameterization.** We parameterize $p(\omega; \boldsymbol{\eta_\omega})$ as a spline flow,

$$p(\omega; \boldsymbol{\eta_\omega}) = p(\zeta)\left|\frac{\partial T(\zeta; \boldsymbol{\eta_\omega})}{\partial \zeta}\right|^{-1} \tag{7}$$

although it could, in principle, be any positive, finite probability distribution. Here, $p(\zeta) \sim \mathcal{N}(0, 1)$ is the base distribution and $\omega = T(\zeta; \boldsymbol{\eta_\omega})$ represents the spline flow transformation followed by a *Softplus* transformation to guarantee $\omega$ to be positive. The flexibility of this flow-based construction lets us capture a broad range of elliptical likelihoods, but we could also specify an appropriate likelihood ourselves. For instance, using a categorical likelihood enables $\mathcal{EP}$ classification, see section 4.3.

**Training objective.** Now, assume that we observe $N$ independent and identically distributed residuals $\epsilon_n = y_n - f_n$ between the observations $\boldsymbol{y}$ and some function, $\boldsymbol{f}$. We are primarily interested in estimating the noise for the purpose of "denoising" the measurements. Hence, we fit an elliptical distribution to the residuals by maximizing the (log) marginal likelihood with respect to the parameters $\boldsymbol{\eta_\omega}$, that is

$$\log p(\boldsymbol{\epsilon}; \boldsymbol{\eta_\omega}) = \sum_{n=1}^{N} \log \int \mathcal{N}(\epsilon_n; 0, T(\zeta; \boldsymbol{\eta_\omega}))\left|\frac{\partial T(\zeta; \boldsymbol{\eta_\omega})}{\partial \zeta}\right|^{-1}p(\zeta)d\zeta. \tag{8}$$

For general mixing distributions this integral is intractable, but we can approximate it using variational inference.

**Variational approximation.** Instead of optimizing the marginal likelihood (8) directly, we approximate the posterior base distribution $p(\zeta|\epsilon_n) \approx q(\zeta_n; \boldsymbol{\varphi}_{\zeta_n})$, where $\boldsymbol{\varphi}_{\zeta_n}$ are variational parameters, and maximize

the corresponding evidence lower bound (ELBO):

$$\mathcal{L}(\boldsymbol{\eta}_\omega, \boldsymbol{\varphi}_{\zeta_1}, \ldots, \boldsymbol{\varphi}_{\zeta_N}) = \sum_{n=1}^{N} \mathbb{E}_{q(\zeta_n; \boldsymbol{\varphi}_{\zeta_n})} \left[ \log \left( \mathcal{N}(\epsilon_n; 0, T(\zeta; \boldsymbol{\eta}_\omega)) \left| \frac{\partial T(\zeta; \boldsymbol{\eta}_\omega)}{\partial \zeta} \right|^{-1} \right) \right] - D_{\mathrm{KL}}\left( q(\zeta_n; \boldsymbol{\varphi}_{\zeta_n}) || p(\zeta) \right). \tag{9}$$

In this variational approximation we have one set of variational parameters $\boldsymbol{\varphi}_{\zeta_n}$ for each observed noise $\epsilon_n$. To reduce complexity, we amortize (see Section 2.4) the variational parameters by letting $\boldsymbol{\varphi}_{\zeta_n} = g(\epsilon_n; \boldsymbol{\gamma}_\zeta)$, which reduces the ELBO to $\mathcal{L}(\boldsymbol{\eta}_\omega, \boldsymbol{\gamma}_\zeta)$. Specifically, we model the variational posterior as a Normal distribution $q(\zeta_n) = \mathcal{N}(\mu_\zeta(\epsilon_n), \sigma_\zeta(\epsilon_n))$ where the variational parameters $\boldsymbol{\varphi}_{\zeta_n}$, namely the mean $\mu_\zeta$ and standard deviation $\sigma_\zeta$, are functions defined by a neural network with parameters $\boldsymbol{\gamma}_\zeta$. The parameters of the normalizing flow and variational posterior are trained jointly by gradient-based optimization of the ELBO, $\nabla_{\boldsymbol{\eta}_\omega, \boldsymbol{\gamma}_\zeta} \mathcal{L}(\boldsymbol{\eta}_\omega, \boldsymbol{\gamma}_\zeta)$. The gradients are estimated using black-box variational inference (Bingham et al., 2019).

Ultimately, we arrive at the likelihood

$$p(\boldsymbol{y}|\boldsymbol{f}) = \prod_{n=1}^{N} \int \mathcal{N}(y_n; f_n, T(\zeta; \boldsymbol{\eta}_\omega)) \left| \frac{\partial T(\zeta; \boldsymbol{\eta}_\omega)}{\partial \zeta} \right|^{-1} p(\zeta) d\zeta. \tag{10}$$

Note that the variational posterior $q(\zeta_n)$ does not appear in this expression—it is only used as an aid for training the mixing distribution (specifically, the parameters $\boldsymbol{\eta}_\omega$).

## 3.2 Prior

Recall that our main objective is to infer the latent *function* $f^* = f(\boldsymbol{x}^*)$ at arbitrary locations $\boldsymbol{x}^* \in \mathbb{R}^D$ given a finite set of noisy observations $\boldsymbol{y}$. In probabilistic machine learning, the mapping $\boldsymbol{y} \mapsto f^*$ is often defined by the posterior predictive distribution

$$p(f^*|\boldsymbol{y}) = \int p(f^*|\boldsymbol{f})p(\boldsymbol{f}|\boldsymbol{y})d\boldsymbol{f}, \tag{11}$$

which turns modeling into a search for suitable choices of $p(f^*|\boldsymbol{f})$ and $p(\boldsymbol{f}|\boldsymbol{y})$. Accordingly, the noise estimation described in the previous section is only done in pursuit of this higher purpose.

**Sparse formulation.** For an elliptical process ($\mathcal{EP}$) we can rewrite the posterior predictive distribution as

$$p(\boldsymbol{f}^*|\boldsymbol{y}) = \int p(\boldsymbol{f}^*|\boldsymbol{f}, \xi) \, p(\boldsymbol{f}, \boldsymbol{u}, \xi|\boldsymbol{y}) d\boldsymbol{f} d\boldsymbol{u} \, d\xi, \tag{12}$$

where we are marginalizing not only over the mixing variable $\xi$ and the function values $\boldsymbol{f}$ (at the given inputs $\boldsymbol{x}$) but also over the function values $\boldsymbol{u}$ at the, so called, $M$ inducing inputs $\boldsymbol{X}_u$. Introducing inducing points lets us derive a *sparse* variational $\mathcal{EP}$—a computationally scalable version of the $\mathcal{EP}$ similar to the sparse variational $\mathcal{GP}$ (Titsias, 2009). We refer to Appendix D for a non-sparse version of the model.

The need for approximation arises because of the intractable second factor, $p(\boldsymbol{f}, \boldsymbol{u}, \xi|\boldsymbol{y})$, in (12). (The first factor, $p(\boldsymbol{f}^*|\boldsymbol{f}, \xi)$, is simply a Normal distribution.)

**Variational approximation.** We make the variational ansatz $p(\boldsymbol{f}, \boldsymbol{u}, \xi|\boldsymbol{y}) \approx p(\boldsymbol{f}|\boldsymbol{u}, \xi)q(\boldsymbol{u}, \xi) \, q(\xi)$, and parameterize this variational posterior as an elliptical distribution. We do so for two reasons: first, this makes the variational posterior similar to the true posterior, and second, we can then use the conditional distribution to make predictions. In full detail, we factorize the posterior as

$$q(\boldsymbol{f}, \boldsymbol{u}, \xi; \boldsymbol{\varphi}) = p(\boldsymbol{f}|\boldsymbol{u}, \xi; \boldsymbol{\eta_f})q(\boldsymbol{u}|\xi; \boldsymbol{\varphi_u})q(\xi; \boldsymbol{\varphi_\xi}), \tag{13}$$

where $\boldsymbol{\varphi} = (\boldsymbol{\varphi_f}, \boldsymbol{\varphi_u}, \boldsymbol{\varphi_\xi})$ are the variational parameters, $q(\boldsymbol{u}|\xi; \boldsymbol{\varphi_u}) = \mathcal{N}(\mathbf{m}, \mathbf{S}\xi)$ is a Gaussian distribution with the variational mixing distribution $\xi \sim q(\xi; \boldsymbol{\varphi_\xi})$. Again, $q(\xi; \boldsymbol{\varphi_\xi})$ could be any positive finite distribution, but we parameterize it with a spline flow.

Note that, because of the conditioning on $\xi$, the first two factors in (13) is a Gaussian conjugate pair in $\boldsymbol{u}$. Thus, marginalization over $\boldsymbol{u}$ results in a Gaussian distribution, for which the marginals of $f_n$ only depends on the corresponding input $\boldsymbol{x}_n$ (Salimbeni et al., 2019):

$$q(f_n|\xi; \boldsymbol{\varphi}) = \mathcal{N}(f_n|\mu_{\boldsymbol{f}}(\boldsymbol{x}_n), \sigma_{\boldsymbol{f}}(\boldsymbol{x}_n)\xi), \tag{14}$$

where

$$\mu_{\boldsymbol{f}}(\boldsymbol{x}_n) = \boldsymbol{k}_n^\top \boldsymbol{K}_{\boldsymbol{uu}}^{-1} \boldsymbol{m}, \tag{15}$$

$$\sigma_{\boldsymbol{f}}(\boldsymbol{x}_n) = k_{nn} - \boldsymbol{k}_n^\top \left( \boldsymbol{K}_{\boldsymbol{uu}}^{-1} - \boldsymbol{K}_{\boldsymbol{uu}}^{-1} \boldsymbol{S} \boldsymbol{K}_{\boldsymbol{uu}}^{-1} \right) \boldsymbol{k}_n, \tag{16}$$

and $\boldsymbol{k}_n = k(\boldsymbol{x}_n, \boldsymbol{X}_{\boldsymbol{u}})$, $k_{nn} = k(\boldsymbol{x}_n, \boldsymbol{x}_n)$, and $\boldsymbol{K}_{\boldsymbol{uu}} = k(\boldsymbol{X}_{\boldsymbol{u}}, \boldsymbol{X}_{\boldsymbol{u}})$.

Predictions on unseen data points, $\boldsymbol{x}^*$, are then computed according to (see Appendix E)

$$p(f^*|\mathbf{y}; \boldsymbol{x}^*) = \mathbb{E}_{q(\xi; \boldsymbol{\varphi}_\xi)} \left[ \mathcal{N}(f^*|\mu_{\boldsymbol{f}}(\boldsymbol{x}^*), \sigma_{\boldsymbol{f}}(\boldsymbol{x}^*)\xi) \right], \tag{17}$$

We consider two distinct methods for training the variational parameters: (VI) variational inference, i.e. maximizing the evidence lower bound to indirectly maximize the marginal likelihood, and (PP) maximum likelihood estimation of the posterior predictive distribution (Jankowiak et al., 2020). Both models are, however, trained by stochastic gradient descent and black-box variational inference (Bingham et al., 2019; Wingate & Weber, 2013; Ranganath et al., 2014).

**VI training.** The marginal likelihood is

$$p(\boldsymbol{y}; \boldsymbol{\eta_f}, \boldsymbol{\eta}_\xi) = \int p(\boldsymbol{y}, \boldsymbol{f}, \boldsymbol{u}, \xi; \boldsymbol{\eta_f}, \boldsymbol{\eta}_\xi) d\boldsymbol{f} d\boldsymbol{u} d\xi = \int p(\boldsymbol{y}|\boldsymbol{f}) p(\boldsymbol{f}|\boldsymbol{u}, \xi; \boldsymbol{\eta_f}) p(\xi; \boldsymbol{\eta}_\xi) d\boldsymbol{f} d\boldsymbol{u} d\xi. \tag{18}$$

However, this integral is intractable—just as it was for the elliptical likelihood—since $p(\xi; \boldsymbol{\eta}_\xi)$ is parameterized by a spline flow. To overcome this we use the same procedure as for the likelihood model and approximate the marginal likelihood with the ELBO

$$\mathcal{L}_{\text{ELBO}}(\boldsymbol{\eta_f}, \boldsymbol{\eta}_\xi, \boldsymbol{\varphi_f}, \boldsymbol{\varphi}_\xi) = \mathbb{E}_{q(\boldsymbol{f}, \xi; \boldsymbol{\varphi})} \left[ \log p(\mathbf{y}|\boldsymbol{f}) \right] - \log D_{\text{KL}} \left( q(\boldsymbol{u}, \xi; \boldsymbol{\varphi}) \,||\, p(\boldsymbol{u}, \xi; \boldsymbol{\eta}) \right)$$
$$= \sum_{n=1}^N \mathbb{E}_{q(f_n, \xi; \boldsymbol{\varphi})} \left[ \log p(y_n|f_n) \right] - \log D_{\text{KL}} \left( q(\boldsymbol{u}, \xi; \boldsymbol{\varphi}) \,||\, p(\boldsymbol{u}, \xi; \boldsymbol{\eta}) \right), \tag{19}$$

Had the likelihood been Gaussian, the expectation $\mathbb{E}_{q(f_n, \xi; \boldsymbol{\varphi})} \left[ \log p(y_n|f_n; \boldsymbol{\eta_f}) \right]$ could have been computed analytically. In our case, however, it is elliptical and we therefore use a Monte Carlo estimate instead. Inserting the elliptical likelihood (10) from the previous section gives

$$\mathcal{L}(\boldsymbol{\eta}, \boldsymbol{\varphi}) = \sum_{n=1}^N \mathbb{E}_{q(f_n, \xi; \boldsymbol{\varphi}) q(\zeta_n; \boldsymbol{\varphi}_{\zeta_n})} \left[ \log \left( \mathcal{N}\left( y_n; f_n, T(\zeta; \boldsymbol{\eta}_\omega) \right) \left| \frac{\partial T(\zeta; \boldsymbol{\eta}_\omega)}{\partial \zeta} \right|^{-1} \right) \right]$$
$$- \sum_{n=1}^N D_{\text{KL}} \left( q(\zeta_n; \boldsymbol{\varphi}_{\zeta_n}) || p(\zeta) \right) - \log D_{\text{KL}} \left( q(\boldsymbol{u}, \xi; \boldsymbol{\varphi}) || p(\boldsymbol{u}, \xi; \boldsymbol{\eta}) \right). \tag{20}$$

**PP training.** To train directly on the predictive posterior of the elliptical process has the effect of moving the posterior distribution $q(f_n|\xi; \boldsymbol{x}_n)$ inside the log (Jankowiak et al., 2020),

$$\mathcal{L}(\boldsymbol{\eta}, \boldsymbol{\varphi}) = \sum_{n=1}^N \mathbb{E}_{q(\zeta_n; \boldsymbol{\varphi}_{\zeta_n}) q(\xi; \boldsymbol{\varphi}_\xi)} \left[ \log \left( \mathcal{N}\left( y_n; \mu_{\boldsymbol{f}}(\boldsymbol{x}_n), \sigma_{\boldsymbol{f}}(\boldsymbol{x}_n)\xi + T(\zeta; \boldsymbol{\eta}_\omega) \right) \left| \frac{\partial T(\zeta; \boldsymbol{\eta}_\omega)}{\partial \zeta} \right|^{-1} \right) \right]$$
$$- \sum_{n=1}^N D_{\text{KL}} \left( q(\zeta_n; \boldsymbol{\varphi}_{\zeta_n}) || p(\zeta) \right) - \log D_{\text{KL}} \left( q(\boldsymbol{u}, \xi; \boldsymbol{\varphi}) || p(\boldsymbol{u}, \xi; \boldsymbol{\eta}) \right] \tag{21}$$

Note that when drawing a single Monte Carlo sample from $q(f_n, \xi; \boldsymbol{\varphi})$ the two methods are equivalent. Similarly, the expectation over $q(\xi; \boldsymbol{\varphi}_\xi)$ can be moved inside the log if a single Monte Carlo sample from $q(\xi; \boldsymbol{\varphi}_\xi)$ is used.

### 3.3 Extension to heteroscedastic noise

We extend the elliptical likelihood by modeling heteroscedastic noise. First, recall from Section 3.1 that we amortized the variational mixing distribution for the elliptical likelihood. Here, we describe how we can model elliptic heteroscedastic noise by letting the parameters $\boldsymbol{\eta}_\omega$ of the mixing distribution of the likelihood depend on the input location.

In heteroscedastic regression, the noise depends on the input location $\boldsymbol{x}_n$. For example, heteroscedastic elliptical noise can be useful in a time series where the noise variance and tail-heaviness change over time. Examples of this can be found in statistical finance (Liu et al., 2020) and robotics (Kersting et al., 2007). To model this, we return to the idea of amortized variational inference and use a neural network with parameters $\gamma_\omega$ to represent the mapping from input location to spline flow parameters, $\boldsymbol{x}_n \mapsto \boldsymbol{\eta}_{\omega_n}$. To train the likelihood (see Section 3.1) we used a variational approximation of the posterior base distribution but kept the same flow as the mixing distribution. As shown later, in Section 4.1, this works well for homoscedastic noise. For heteroscedastic noise, on the other hand, we got better results by instead keeping the base distribution fixed and learning a different spline flow with parameters dependent on both input location and noise, $\boldsymbol{\varphi}_{\omega_n} = g(\boldsymbol{x}_n, \epsilon_n; \tilde{\boldsymbol{\gamma}}_\omega)$.

We train the model by minimizing the ELBO

$$\mathcal{L}(\boldsymbol{\gamma}_\omega, \tilde{\boldsymbol{\gamma}}_\omega) = \sum_{n=1}^{N} \mathbb{E}_{\zeta_n \sim q(\boldsymbol{\eta}_n; \boldsymbol{\varphi}_{\omega_n})} \left[ \log p(\epsilon_n | \omega_n) \right] - D_{\mathrm{KL}} \left( q(\omega_n; \boldsymbol{\varphi}_{\omega_n}) \,||\, p(\omega_n; \boldsymbol{\eta}_{\omega_n}) \right). \tag{22}$$

This model can be extended by including additional inputs to the spline flow.

### 3.4 Extension to multi-path data

Here, we look at data with multiple independent realizations from the same $\mathcal{EP}$ prior, called sample paths, see Figure 4. For example, it could be multiple time series generated by an underlying physical process like temperature or pressure. Suppose we have $M$ sample paths such that every pair $(\boldsymbol{y}^m, \boldsymbol{X}^m)$ represent one of the trajectories. We create a model where the sample paths share the same $\mathcal{EP}$ prior, i.e. have the same mixing distribution and kernel, but where each sample path has its own approximate posterior

$$\begin{aligned} q(\xi; \boldsymbol{\varphi}_\xi^m) &\approx p(\xi | \boldsymbol{y}^m; \boldsymbol{\eta}_\xi), \\ q(\boldsymbol{u}; \boldsymbol{\varphi}_{\boldsymbol{u}}^m) &\approx p(\boldsymbol{u} | \boldsymbol{y}^m; \boldsymbol{\eta}_{\boldsymbol{u}}). \end{aligned} \tag{23}$$

By sharing the prior, the hope is that the model will be less prone to overfitting. Also, the final $\mathcal{EP}$ prior represents a unified representation of all sample paths, which may provide additional insight into the underlying process.

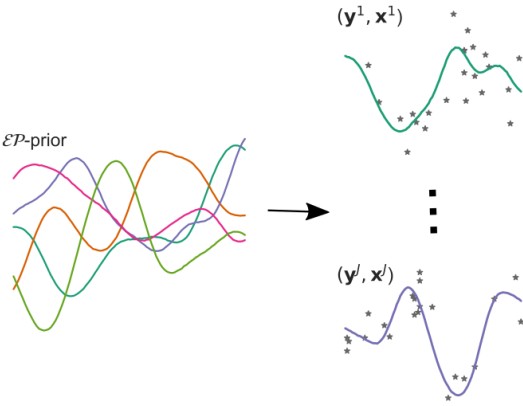

Figure 4: Illustration of multi-path $\mathcal{EP}$-data, where the data includes multiple draws from one single $\mathcal{EP}$-prior.

The main challenge of this model is that each path has a separate tuple of variational parameters $(\boldsymbol{\eta}_{\xi,m}, \boldsymbol{m}^m, \boldsymbol{S}^m)$, which can be problematic if there are many of them. We, again, resolve this by amortizing the variational parameters:

$$\begin{aligned} \boldsymbol{\varphi}_\xi^m &= g(\boldsymbol{y}^m, \boldsymbol{X}^m; \gamma_\xi), \\ \boldsymbol{m}^m, \boldsymbol{S}^m &= g(\boldsymbol{y}^m, \boldsymbol{X}^m; \gamma_{\boldsymbol{u}}). \end{aligned} \tag{24}$$

The functions $g(\,\cdot\,; \gamma_\xi)$ and $g(\,\cdot\,; \gamma_{\boldsymbol{u}})$ are parameterized by neural networks with parameters $\gamma_\xi$ and $\gamma_{\boldsymbol{u}}$ in a similar fashion as in Jafrasteh et al. (2021).

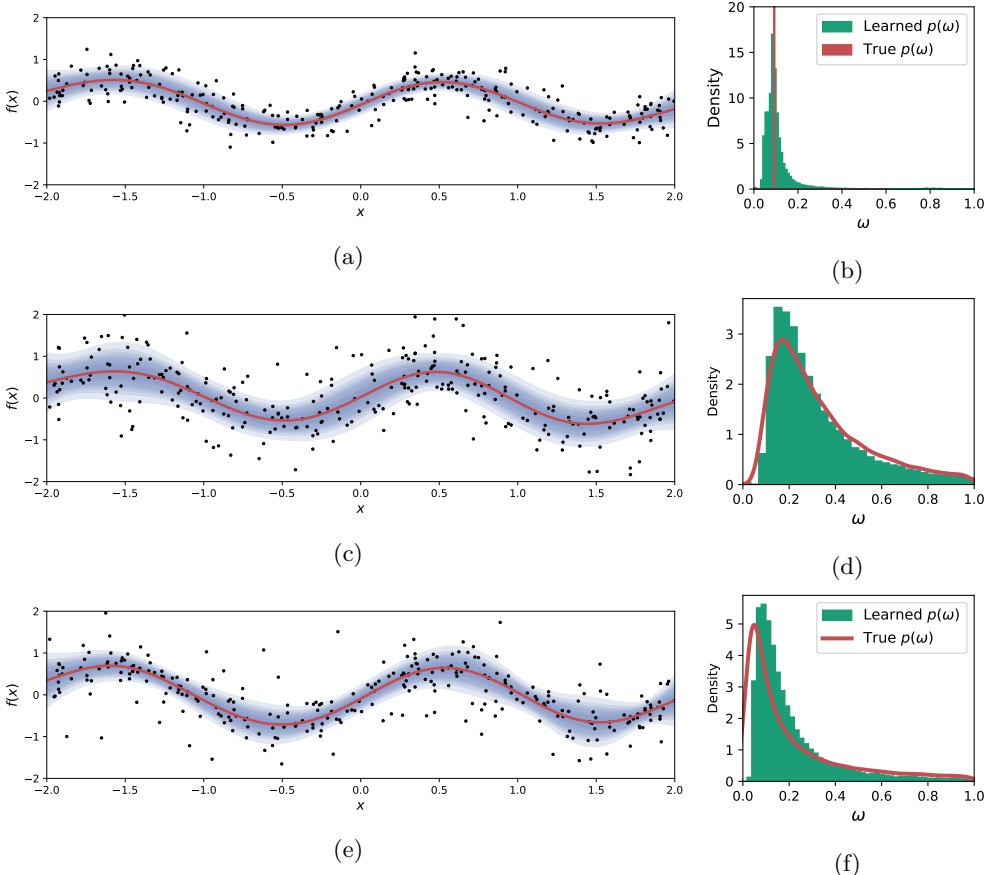

Figure 5: The posterior predictive distribution when using an $\mathcal{GP}$ with elliptical noise, modeled with a spline flow. The histograms show the learned and the true noise mixing distribution.

## 4 Experiments

We examine the variational elliptical processes using five different experiments. In the **first** experiment, we investigate how well the elliptical likelihood (Section 3.1) recovers known elliptical noise in synthetic data. In the **second** experiment, we investigate the benefits of using the sparse $\mathcal{EP}$ compared to the sparse $\mathcal{GP}$ for regression on standard benchmarks. In the **third** experiment, we examine if using a $\mathcal{EP}$ is beneficial in classification tasks. In the **fourth** experiment, we investigate the amortized elliptical processes described in Section 3.3 to model heteroscedastic noise. In the **fifth** and last experiment we illustrate how we can use an amortized multi-path $\mathcal{EP}$ ( Section 3.3) on a dataset with multiple similar trajectories.

**Implementation.** The mixing distribution of the variational $\mathcal{EP}$ uses a linear rational spline flow, where we transform the likelihood flow $p(\omega)$ using *Softplus* and the posterior flow $p(\xi)$ using a *Softmax* to ensure that it is bounded from below and positive. We use a squared exponential kernel with independent length scales in all experiments. See Appendix F for further implementation details. The code from the experiments will be published on GitHub if the paper is accepted, with a link added here.

### 4.1 Noise identification

To examine how well the elliptical likelihood, described in Section 3.1, captures different types of elliptical noises, we created three equal synthetic datasets, each with $N = 300$ data points, by using the function $f_n = \sin(3\boldsymbol{x}_n)/2$, where $\boldsymbol{x} \in \mathbb{R}$ is uniformly sampled, $\boldsymbol{x}_n \sim U(-2, 2)$. Each of the dataset has its own independent elliptical noise $\epsilon_n$, which are andomly sampled and added to the function, $y_n = f_n + \epsilon_n$. For

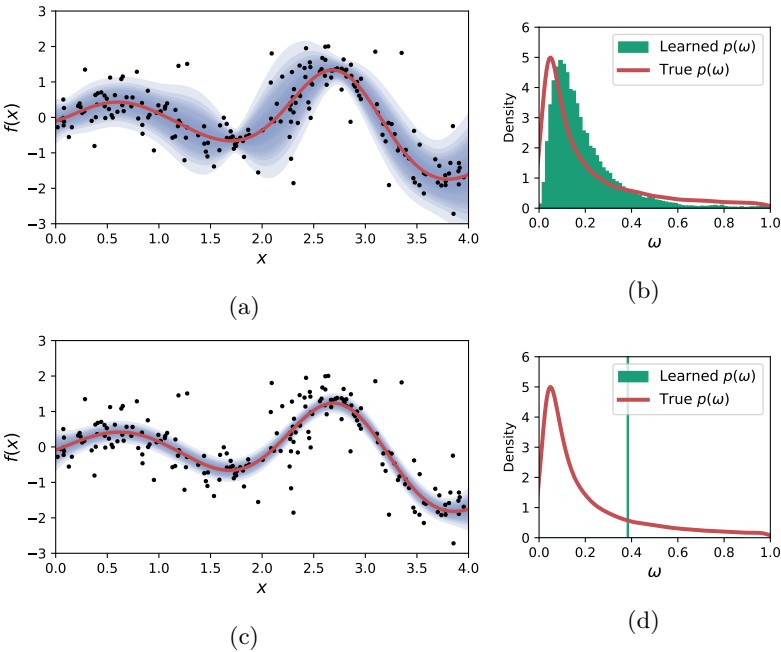

Figure 6: The predictive distribution of the latent function together with the 99 % credibility interval when using (**top row**) an $\mathcal{GP}$ with elliptical noise, modeled with a spline flow and a (**bottom row**) a $\mathcal{GP}$ with Gaussian noise. The histograms show the learned and the true noise mixing distribution.

each dataset, we trained a sparse variational $\mathcal{GP}$ with a variational elliptical likelihood. For the loss, we used the parametric posterior (21).

Figure 5 illustrates the results from the experiments. The histograms illustrated the trained mixing distribution $p(\omega; \boldsymbol{\eta}_\omega)$ which we compare to the actual mixing distribution (the red curve) from which the noise $\epsilon_n$ originated. The learned distribution follows the shape of the actual mixing distribution quite well, which indicates that it is possible to learn a noise mixing distribution. The predictive posterior (plotted in Figure 21). The figure also presents the predictive posterior of the final models, demonstrating that the models learned suitable kernel parameters simultaneously as they learned the likelihood mixing distribution.

Figure 6 compares the final mixing distribution using an elliptical likelihood and a Gaussian one. We see that the Gaussian likelihood matches the heavier-tailed mixing distribution with a variance ($\omega = 0.4$) that is too wide. This results in a latent function confidence interval that is extremely narrow.

## 4.2 Regression

We investigated the effects of the elliptical process and the elliptical noise by running experiments on several datasets from the UCI repository (Dua & Graff, 2017). The models we investigate, summarized in Table 1, all use a $\mathcal{GP}$ prior, but for some models, we use an approximated $\mathcal{EP}$ posterior instead of the regular $\mathcal{GP}$ posterior. We compare our model to the sparse variational $\mathcal{GP}$ described in Hensman et al. (2013b), which we call VI-$\mathcal{GP}$-$\mathcal{GP}$ and the parametric $\mathcal{GP}$ described in Jankowiak et al. (2020), which we call PP-$\mathcal{GP}$-$\mathcal{GP}$. We also compare the models to an exact $\mathcal{GP}$ for all but the two largest datasets.

Figure 7 summarizes the results from the experiment by plotting the mean and standard deviation from the outcome of ten randomly sampled training validation and test data points. The figures show a hold-out test set's mean squared error (MSE) and the negative test log-likelihood (LL). See Appendix G for more details.

An elliptic likelihood gives a lower negative log-likelihood than a Gaussian likelihood on most datasets. However, the advantage of an elliptical likelihood on the three smaller datasets is small at best. We hypothesize

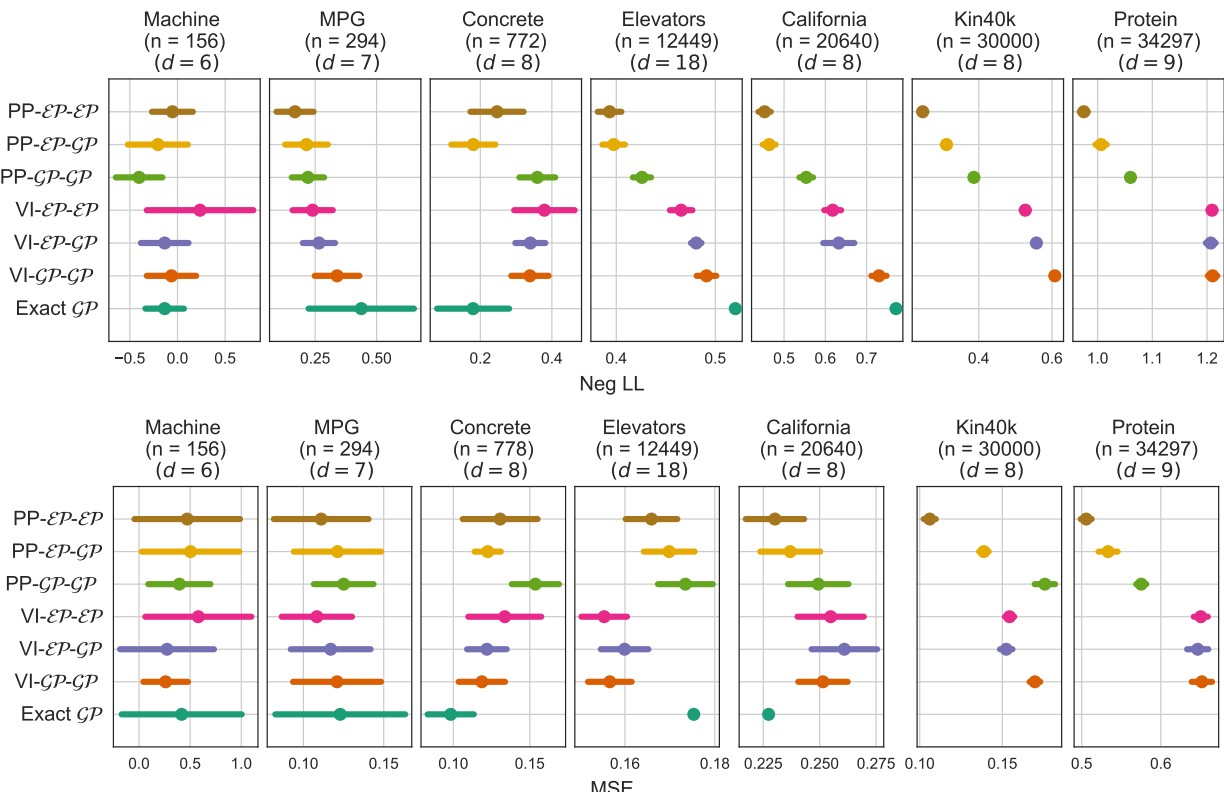

Figure 7: Predictive negative log-likelihood (LL) and mean-squared error (MSE) on the hold out sets from the experiments (smaller is better). We show the average of the ten folds and standard deviation as a line.

Table 1: The different types of models we train on the regression datasets.

| Name | Approx | Loss | Likelihood | Posterior |
|------|--------|------|------------|-----------|
| Exact $\mathcal{GP}$ | Exact | Marginal likelihood | Gaussian | Gaussian |
| VI-$\mathcal{GP}$-$\mathcal{GP}$ | Variational | ELBO | Gaussian | Gaussian |
| VI-$\mathcal{EP}$-$\mathcal{GP}$ | Variational | ELBO | Elliptic | Gaussian |
| VI-$\mathcal{EP}$-$\mathcal{EP}$ | Variational | ELBO | Elliptic | Elliptic |
| PP-$\mathcal{GP}$-$\mathcal{GP}$ | Variational | Parametric | Gaussian | Gaussian |
| PP-$\mathcal{EP}$-$\mathcal{GP}$ | Variational | Parametric | Elliptic | Gaussian |
| PP-$\mathcal{EP}$-$\mathcal{EP}$ | Variational | Parametric | Elliptic | Elliptic |

that the elliptic likelihood may be too flexible and overfits the training data. Potentially, a less flexible mixing distribution combined with stronger regularization might improve performance.

On the larger datasets the elliptic posterior yields lower negative log-likelihoods compared to a Gaussian posterior, even though the extra benefit from only the elliptic likelihood is marginal. Theoretically, a Gaussian prior combined with an elliptical likelihood should yield an elliptical poster. However, finding the correct posterior during training might be challenging, which could be why we only see the benefit for the largest datasets.

We notice that for the majority of the datasets, we get a lower negative predictive log-likelihood when we train the predictive log-likelihood directly. This is true for both the $\mathcal{GP}$ and the $\mathcal{EP}$ models. However, the improvement is not as clear when considering the mean square error, even though we see a considerably decreased MSE on the three largest datasets.

### 4.3 Binary classification

To evaluate the $\mathcal{EP}$ on classifications tasks, we perform variational $\mathcal{EP}$ and $\mathcal{GP}$ classification by simply replacing the likelihood with a binary one. To derive the expectation in Equation 19 we first sample $f_n \sim \mathcal{N}(f_n|\mu_{\boldsymbol{f}}(\boldsymbol{x}_n), \sigma_{\boldsymbol{f}}(\boldsymbol{x}_n)\xi)$ and then derive the likelihood as $\mathrm{Ber}(\mathrm{Sigmoid}(f_i))$.

This realization is interesting since here, we do not have a likelihood that captures the noise in the data, but instead, the process itself has to do it. Therefore, we can indicate the value of the elliptical process itself without the elliptical noise. We compare two sparse $\mathcal{EP}$ models with a sparse $\mathcal{GP}$ model using 20 inducing points. The two $\mathcal{EP}$s differs in the prior mixing distribution. We used a $\mathcal{GP}$ prior and a $\mathcal{EP}$ posterior for the first model. For the second model, we insted replace the $\mathcal{GP}$ prior to an elliptical one. We can see the trainable prior mixing distribution as using a continuously scaled mixture of Gaussian processes, which can be more expressive than a single $\mathcal{GP}$.

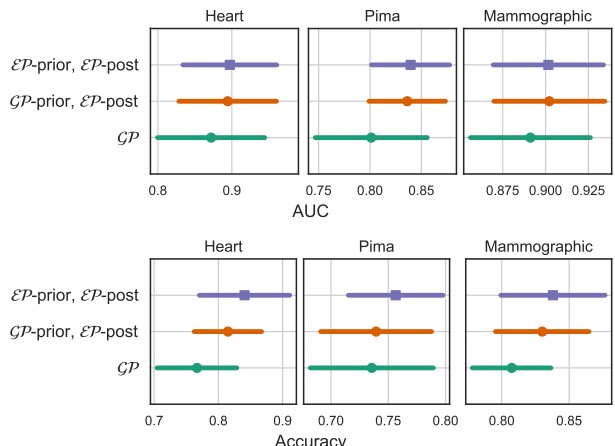

Figure 8: The classification AUC (Area Under the Curve) and accuracy score from the ten-fold cross validation (higher is better).

We trained the models on three classification datasets, described in Appendix I. The results from a ten-fold cross-validation is presented in Figure 8. From the area under the curve (AUC) score, we see that the $\mathcal{EP}$ prior separates the two classes better. It seems that the variational elliptical distribution mainly contributes to the higher AUC score. Training the mixing distribution of the $\mathcal{EP}$ prior did not improve the score.

### 4.4 Elliptic heteroskedastic noise

In this experiment, we aimed to learn heteroscedastic noise as described in Section 3.3 on a synthetic dataset of 150 samples, see Figure 9. We created the dataset using the function $f(x) = \sin(5x) + x$. We then added Student's $t$ noise, $\epsilon(x) \sim St(\nu(x), \sigma(x))$, where we decreased the noise scale by $\nu(x) = 25 - 11|x + 1|^{0.9}$, and the increased the standard deviation by $\sigma(x) = 0.5|x+1|^{1.6} + 0.001$. We used a variational sparse $\mathcal{GP}$ with heteroscedastic noise as described in Section 3.3.

We used six bins for the prior mixing distribution and eight bins for the posterior mixing distribution, which resulted in 19 and 35 parameters to predict, respectively. We had more bins for the posterior mixing distribution since we wanted the approximate posterior to be as flexible as possible to fit the true posterior.

The results from the experiments are depicted in Figure 9 and show that the model was able to capture the varying noise, both in term of the scale and the increasing heaviness of the tail. A single spike in the mixing distribution indicates that the noise is Gaussian, and the *wider* the mixing distribution is, the heavier tailed the noise is.

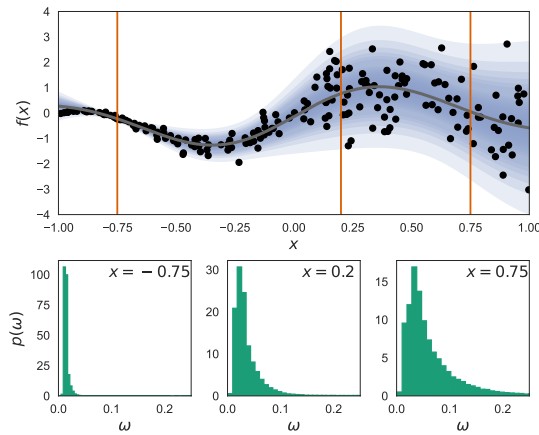

Figure 9: The result form training a $\mathcal{GP}$ process with heteroscedastic elliptical noise on a synthetic dataset. The histogram shows the noise resulting mixing distributions ad different $\boldsymbol{x}_n$.

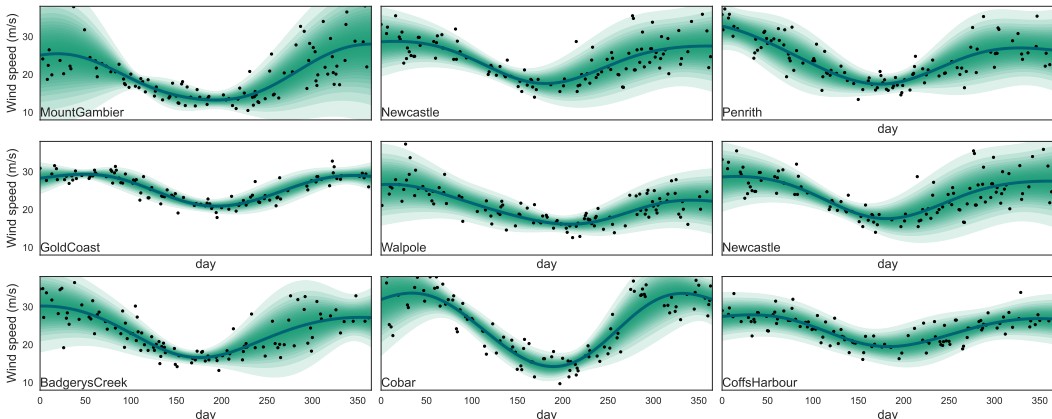

Figure 11: Posterior predictive distributions of the wind speed during one year at nine different locations in Australia. They all share the same $\mathcal{EP}$ prior but have data-dependent posterior predictive distributions.

### 4.5 Multi-path data

Here we experimented with multi-path data, as described in Section 3.4. The dataset, (Young & Young, 2020), contains daily temperature observations from $J = 49$ different locations in Australia in 2015. We randomly divided trajectories (time series) corresponding to different locations into training and test sets. We used a multi-path $\mathcal{EP}$-process (Section 3.4) since we assumed that temperature trajectories at different locations still have an underlying similarity and thus correspond to sample paths from the same $\mathcal{EP}$-prior. Furthermore, we used the same elliptical likelihood for all trajectories.

The variational mixing distribution parameters $\boldsymbol{\varphi}_\xi^m$ are amortized by a dense neural networks with two hidden layers, each with 512 hidden units. The parameters $\boldsymbol{\mu}^m$ and $\boldsymbol{\Sigma}^m$ are amortized by a dense neural networks with two hidden layers, with 512 and 1024 hidden units for $\boldsymbol{\mu}^m$ and $\boldsymbol{\Sigma}^m$, respectively.

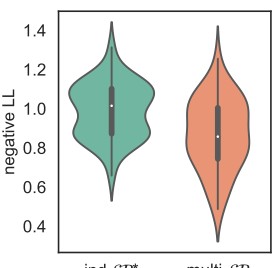

Figure 10: Negative predictive log likelihood (LL) of the multi-$\mathcal{EP}$ compare with modelling every trajectory individually (ind $\mathcal{EP}$).

Figure 11 illustrate the resulting posterior distribution $q(\boldsymbol{f}^m; \boldsymbol{\varphi})$ for a some of the sample paths. We compare the negative log likelihood when training all trajectories individually (Figure 10) and see an see a decrease in negative log likelihood when sharing the $\mathcal{EP}$ prior.

## 5 Related work

In general, attempts at modeling heavy-tailed stochastic processes modify either the likelihood or the stochastic process prior—rarely both. Approximate inference is typically needed when going beyond Gaussian likelihoods (Neal, 1997; Jylänki et al., 2011), e.g., for robust regression, but approximations that preserve analytical tractability have been proposed (Shah et al., 2014).

Ma et al. (2019) describes a class of stochastic processes where the finite-dimensional distributions are only defined implicitly as a parameterized transformation of some base distribution, thereby generalizing earlier work on warped Gaussian processes (Snelson et al., 2004; Rios & Tobar, 2019). However, the price of this generality is that standard variational inference is no longer possible. Based on an assumption of a Gaussian likelihood, they describe an alternative based on the wake-sleep algorithm by Hinton et al. (1995).

Other attempts at creating more expressive $\mathcal{GP}$ priors include Maroñas et al. (2021), who used a $\mathcal{GP}$ in combination with a normalizing flow, and Luo & Sun (2017), who used a discrete mixture of Gaussian processes. Similar ideas combining mixtures and normalizing flows have also been proposed to create

more expressive likelihoods (Abdelhamed et al., 2019; Daemi et al., 2019; Winkler et al., 2019; Rivero & Dvorkin, 2020) and variational posteriors (Nguyen & Bonilla, 2014). Non-stationary extensions of Gaussian processes, such as when modeling heteroscedastic noise, are quite rare but the mixture model of Li et al. (2021) and the variational model of Lázaro-Gredilla & Titsias (2011) are two examples.

In the statistics literature, it is well-known that the elliptical processes can be defined as scale-mixtures of Gaussian processes (Huang & Cambanis, 1979; O'Hagan, 1991; O'Hagan et al., 1999). However, unlike in machine learning, little emphasis is placed on building the models from data (i.e., training). These models have found applications in environmental statistics because of the field's inherent interest in modeling spatial extremes (Davison et al., 2012). Several works take the mixing distribution as the starting point, like us, and make localized predictions of quantiles (Maume-Deschamps et al., 2017) or other tail-risk measures (Opitz, 2016).

## 6 Conclusions

The Gaussian distribution is the default choice in statistical modeling for good reasons. Even so, far from everything is Gaussian—casually pretending it is, comes at a risk. The elliptical distribution offers a computationally tractable alternative that can capture heavy-tailed distributions. The same reasoning applies when comparing the Gaussian process to the elliptical process. We believe that a sensible approach in many applications would be to start from the weaker assumptions of the elliptical process and let the data decide whether the evidence supports gaussianity.

We constructed the elliptical processes as a scale mixture of Gaussian distributions. By parameterizing the mixing distribution using a normalizing flow, we show how a corresponding elliptical process can be trained using variational inference. The variational approximation we propose enables us to capture heavy-tailed posteriors and makes it straightforward to create a sparse variational elliptical process that scales to large datasets.

We performed experiments on regression and classification. In addition, we compared the elliptical processes with the Gaussian process. Our experiments show that the elliptical process achieves a lower predictive log likelihood on the majority of the datasets, in particular the larger ones ($n > 10000$).

The added flexibility of the elliptical processes could benefit a range of applications, both classical and new. However, advanced statistical models are not a cure-all, and one needs to avoid over-reliance on such models, especially in safety-critical applications.

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

## A The elliptical distribution

The Gaussian distribution—the basic building block of Gaussian processes—has several attractive properties that we wish the elliptical process to inherit, namely (i) closure under marginalization, (ii) closure under conditioning, and (iii) straightforward sampling. This leads us to consider the family of *consistent* elliptical distributions. Following Kano (1994), we say that a family of elliptical distributions $\{p(u(\mathbf{y}_N); \boldsymbol{\eta}) \,|\, N \in \mathbb{N}\}$ is consistent if and only if

$$\int_{-\infty}^{\infty} p\left(u(\mathbf{y}_{N+1}); \boldsymbol{\eta}\right) dy_{N+1} = p\left(u(\mathbf{y}_N); \boldsymbol{\eta}\right). \tag{25}$$

In other words, a consistent elliptical distribution is closed under marginalization.

Far from all elliptical distributions are consistent, but the complete characterization of those that are is provided by the following theorem (Kano, 1994).

**Theorem 1** *An elliptical distribution is consistent if and only if it originates from the integral*

$$p(u; \boldsymbol{\eta}) = |\boldsymbol{\Sigma}|^{-\frac{1}{2}} \int_0^{\infty} \left(\frac{1}{\xi 2\pi}\right)^{\frac{N}{2}} e^{\frac{-u}{2\xi}} p(\xi; \boldsymbol{\eta}_\xi) d\xi, \tag{26}$$

*where $\xi$ is a mixing variable with the corresponding, strictly positive finite, mixing distribution $p(\xi; \boldsymbol{\eta})$, that is independent of $N$.*

This shows that consistent elliptical distributions $p(u; \boldsymbol{\eta})$ are scale-mixtures of Gaussian distributions, with a mixing variable $\xi \sim p(\xi; \boldsymbol{\eta})$. Note that any mixing distribution fulfilling Theorem 1 can be used to define a consistent elliptical process. We recover the Gaussian distribution if the mixing distribution is a Dirac delta function and the Student's $t$ distribution if it is a scaled chi-square distribution.

If $p(u; \boldsymbol{\eta})$ is a scale-mixture of normal distributions, it has the stochastic representation

$$\mathbf{Y}|\,\xi \sim \mathcal{N}(\boldsymbol{\mu}, \boldsymbol{\Sigma}\xi), \quad \xi \sim p(\xi; \boldsymbol{\eta}). \tag{27}$$

By using the following representation of the elliptical distribution,

$$\mathbf{Y} = \boldsymbol{\mu} + \boldsymbol{\Sigma}^{1/2}\mathbf{Z}\xi^{1/2}, \tag{28}$$

where $\mathbf{Z}$ follows the standard normal distribution, we get the mean

$$\mathbb{E}[\mathbf{Y}] = \boldsymbol{\mu} + \boldsymbol{\Sigma}^{1/2}\mathbb{E}\left[\mathbf{Z}\right]\,\mathbb{E}[\xi^{1/2}] = \boldsymbol{\mu} \tag{29}$$

and the covariance

$$\begin{aligned}
\mathrm{Cov}(\mathbf{Y}) &= \mathbb{E}\left[(\mathbf{Y} - \boldsymbol{\mu})(\mathbf{Y} - \boldsymbol{\mu})^\top\right] \\
&= \mathbb{E}\left[(\boldsymbol{\Sigma}^{1/2}\mathbf{Z}\sqrt{\xi})(\boldsymbol{\Sigma}^{1/2}\mathbf{Z}\sqrt{\xi})^\top\right] \\
&= \mathbb{E}\left[\xi\boldsymbol{\Sigma}^{1/2}\mathbf{Z}\mathbf{Z}^\top(\boldsymbol{\Sigma}^{1/2})^\top\right] \\
&= \mathbb{E}\left[\xi\right]\boldsymbol{\Sigma}.
\end{aligned} \tag{30}$$

The variance is a scale factor of the scale matrix $\boldsymbol{\Sigma}$. To get the variance we have to derive $\mathbb{E}\left[\xi\right]$. Note that if $\xi$ follows the inverse chi-square distribution, $E[\xi] = \nu/(\nu - 2)$. We recognize form the Student's $t$ distribution, where $\mathrm{Cov}(\mathbf{Y}) = \nu/(\nu - 2)\boldsymbol{\Sigma}$.

## B Conditional distribution

To use the $\mathcal{EP}$ for predictions, we need the conditional mean and covariance of the corresponding elliptical distribution, which are derived next. We partition the data as $\mathbf{y} = [\mathbf{y}_1, \mathbf{y}_2]$, where $\mathbf{y}_1$ is the $N_1$ observed data points, $\mathbf{y}_2$ is the $N_2$ data points to predict, and $N_1 + N_2 = N$. We have the following result:

**Proposition 1** *If the data* $\mathbf{y} = [\mathbf{y}_1, \mathbf{y}_2]$ *originate from the consistent elliptical distribution in (3), the conditional distribution originates from the distribution*

$$p_{\mathbf{y_2}|u_1}(\mathbf{y}_2) = \frac{c_{N_1,\boldsymbol{\eta}}}{\left|\boldsymbol{\Sigma}_{22|1}\right|^{\frac{1}{2}} (2\pi)^{\frac{N_2}{2}}} \int_0^\infty \xi^{-\frac{n}{2}} e^{-(u_{2|1}+u_1)\frac{1}{2\xi}} \, p(\xi;\,\boldsymbol{\eta})d\xi \tag{31}$$

*with the conditional mean* $\mathbb{E}[\boldsymbol{y_2}|\boldsymbol{y}_1] = \boldsymbol{\mu}_{2|1}$ *and the conditional covariance*

$$\text{Cov}[\boldsymbol{Y}_2|\boldsymbol{Y}_1 = \boldsymbol{y}_2] = \mathbb{E}[\hat{\xi}]\boldsymbol{\Sigma}_{22|1}, \quad \hat{\xi} \sim \xi|\mathbf{y}_1, \tag{32}$$

*where* $u_1 = (\mathbf{y}_1 - \boldsymbol{\mu}_1)^\top \boldsymbol{\Sigma}_{11}^{-1}(\mathbf{y}_1 - \boldsymbol{\mu}_1)$, $u_{2|1} = (\mathbf{y}_2 - \boldsymbol{\mu}_{2|1})^\top \boldsymbol{\Sigma}_{22|1}^{-1}(\mathbf{y}_2 - \boldsymbol{\mu}_{2|1})$, *and* $c_{N_1,\boldsymbol{\eta}}$ *is a normalization constant. The conditional scale matrix* $\boldsymbol{\Sigma}_{22|1}$ *and the conditional mean vector* $\boldsymbol{\mu}_{2|1}$ *are the same as the mean and the covariance matrix for a Gaussian distribution. The proof is derived in Appendix B.*

The conditional distribution is guaranteed to be a consistent elliptical distribution but not necessarily the same as the original one—the shape depends on the training samples. (Recall that consistency only concerns the marginal distribution.) To prove Proposition 1, we partition the data $\mathbf{y}$ as $[\mathbf{y}_1, \mathbf{y}_2]$, so $n_1$ data points belong to $\mathbf{y}_1$, $n_2$ data points belong to $\mathbf{y}_2$ and $n_1 + n_2 = n$.

**Proof of proposition 1.** The joint distribution of $[\mathbf{y}_1, \mathbf{y}_2]$ is $p(\mathbf{y}_1, \mathbf{y}_2|\xi)p(\xi; \boldsymbol{\eta})$ and the conditional distribution of $\mathbf{y}_2$, given $\mathbf{y}_1$ is $p(\mathbf{y}_2|\mathbf{y}_1, \xi)p(\xi|\mathbf{y}_1 M; \boldsymbol{\eta})$.

For a given $\xi$, $p(\mathbf{y}_2|\mathbf{y}_1, \xi)$ is the conditional normal distribution and so

$$p(\mathbf{y}_2|\mathbf{y}_1, \xi) \sim \mathcal{N}(\boldsymbol{\mu}_{2|1}, \Sigma_{22|1}\hat{\xi}), \quad \hat{\xi} \sim p(\xi|\mathbf{y}_1; \boldsymbol{\eta}) \tag{33}$$

where,

$$\boldsymbol{\mu}_{2|1} = \boldsymbol{\mu}_2 + \Sigma_{21}\Sigma_{11}^{-1}(\mathbf{y}_1 - \boldsymbol{\mu}_1) \tag{34}$$

$$\Sigma_{22|1} = \Sigma_{22} - \Sigma_{21}\Sigma_{11}^{-1}\Sigma_{21}, \tag{35}$$

the same as for the conditional Gaussian distribution. We obtain the conditional distribution $p(\xi|\mathbf{y}_1; \boldsymbol{\eta})$ by remembering that

$$p(\mathbf{y}_1|\xi) \sim \mathcal{N}(\boldsymbol{\mu}_1, \Sigma_{11}\xi). \tag{36}$$

Using Bayes' Theorem we get

$$p(\xi|\mathbf{y}_1; \boldsymbol{\eta}) \propto p(\mathbf{y}_1|\xi)p(\xi; \boldsymbol{\eta})$$

$$\propto |\Sigma_{11}\xi|^{-1/2} \exp\left\{-\frac{u_1}{2\xi}\right\} p(\xi; \boldsymbol{\eta})$$

$$\propto \xi^{-N_1/2} \exp\left\{-\xi\frac{u_1}{2}\right\} p(\xi; \boldsymbol{\eta}). \tag{37}$$

Recall that $u_1 = (\mathbf{y} - \boldsymbol{\mu}_1)^\top \boldsymbol{\Sigma}_{11}^{-1}(\mathbf{y} - \boldsymbol{\mu}_1))$. We normalize the distribution by

$$c_{N_1,\boldsymbol{\eta}}^{-1} = \int_0^\infty \xi^{-N_1/2} \exp\left\{-\frac{u_1}{2\xi}\right\} p(\xi; \boldsymbol{\eta})d\xi \tag{38}$$

The conditional mixing distribution is

$$p(\xi|\mathbf{y}_1; \boldsymbol{\eta}) = c_{N_1,\boldsymbol{\eta}}\xi^{-N_1/2} \exp\left\{-\frac{u_1}{2\xi}\right\} p(\xi; \boldsymbol{\eta}) \tag{39}$$

The conditional distribution of $\mathbf{y}_2$ given $\mathbf{y}_1$ is derived by using the consistency formula

$$p(\mathbf{y}_2|\mathbf{y}_1) = \frac{1}{|\boldsymbol{\Sigma}_{22|1}|^{1/2}(2\pi)^{N_2/2}} \int_0^\infty \xi^{-N_2/2} \exp{-\frac{u_{2|1}}{2\xi}} p(\xi|\mathbf{y}_1)d\xi, \tag{40}$$

where $u_{2|1} = (\mathbf{y}_2 - \boldsymbol{\mu}_{2|1})^\top \Sigma_{22|1}^{-1}(\mathbf{y}_2 - \boldsymbol{\mu}_{2|1})$. Using (39) we get

$$p(\mathbf{y}_2|\mathbf{y}_1) = \frac{c_{N_1,\boldsymbol{\eta}}}{|\boldsymbol{\Sigma}_{22|1}|^{1/2}(2\pi)^{N_2/2}} \int_0^\infty \xi^{-n/2} e^{-(u_{2|1}+u_1)/(2\xi)} p(\xi; \boldsymbol{\eta})d\xi \tag{41}$$

## C   Derivation of the confidence regions of the elliptical process

We derive the confidence region of the elliptical process, by using Monte Carlo approximation of the integral, as

$$p(-z\sigma < x < z\sigma) = \frac{1}{\sigma\sqrt{2\pi}} \int_{-z\sigma}^{z\sigma} \int_0^\infty \xi^{-1/2} e^{-x^2/(\xi 2\sigma^2)} p(\xi) d\xi dx \tag{42}$$

$$= \frac{1}{\sigma\sqrt{2\pi}} \int_{-z\sigma}^{z\sigma} \frac{1}{m} \sum_{i=1}^m \xi_i^{-1/2} e^{-x^2/(2\xi_i\sigma^2)} dx \tag{43}$$

$$= \frac{1}{\sigma m\sqrt{2\pi}} \sum_{i=1}^m \xi_i^{-1/2} \int_{-z\sigma}^{z\sigma} e^{-x^2/(2\xi_i\sigma^2)} dx \tag{44}$$

$$= \frac{2}{m\sqrt{\pi}} \sum_{i=1}^m \int_0^{\frac{z}{\sqrt{2\xi_i}}} e^{-u^2} du \tag{45}$$

$$= \frac{1}{m} \sum_{i=1}^m \mathrm{erf}\left(\frac{z}{\sqrt{2\xi_i}}\right) \tag{46}$$

For every mixing distribution we can derive the confidence of the prediction. It is the number of samples m we take that decides the accuracy of the confidence.

## D   Training the elliptical process

For a Gaussian process the posterior of the latent variables $\boldsymbol{f}$ is

$$p(\boldsymbol{f}|\mathbf{y}) \propto p(\mathbf{y}|\boldsymbol{f})p(\boldsymbol{f}). \tag{47}$$

Here, the prior $p(\boldsymbol{f}|\mathbf{x}) \sim \mathcal{N}(0, K)$, is a Gaussian process with kernel $K$ and the likelihood $p(\mathbf{y}|\mathbf{x}, \boldsymbol{f}) \sim \mathcal{N}(\boldsymbol{f}, \sigma^2\mathbf{I})$ is Gaussian. The posterior derives to

$$p(\boldsymbol{f}|\mathbf{y}) \sim \mathcal{N}\left(\boldsymbol{f}|K\left(K + \sigma^2 I\right)^{-1}\mathbf{y}, \left(\mathbf{K}^{-1} + \sigma^{-2}\mathbf{I}\right)^{-1}\right) \tag{48}$$

and we can derive the predictive distribution of an arbitrary input location $x^*$ by

$$p(f^*|\mathbf{y}) = \int p(f_*|\boldsymbol{f})p(\boldsymbol{f}|\mathbf{y})d\boldsymbol{f}, \tag{49}$$

where $p(f_*|\boldsymbol{f}, \mathbf{x}, \mathbf{x}_*)$ is the conditional distribution, which is again Gaussian with

$$\mathcal{N}\left(f_*|\mathbf{k}_*^\top(\mathbf{k} + \sigma^2\mathbf{I})^{-1}\mathbf{y}, k_{**} - \mathbf{k}_*^\top(\mathbf{K} + \sigma^2\mathbf{I})^{-1}\mathbf{k}_*\right). \tag{50}$$

We want to derive the predictive distribution for the elliptical process, but the problem is that the posterior is intractable. In order to get a tractable posterior, we train the model using variational inference, where we approximate the intractable posterior with a tractable one,

$$p(\boldsymbol{f}, \xi, \omega|\mathbf{y}; \boldsymbol{\eta}) \approx q(\boldsymbol{f}, \xi, \boldsymbol{\omega}; \boldsymbol{\varphi}) = q(\boldsymbol{f}|\xi; \boldsymbol{\varphi_f})q(\xi; \boldsymbol{\varphi_\xi}). \tag{51}$$

Here, $q(\boldsymbol{f}|\xi; \boldsymbol{\varphi}) \sim \mathcal{N}(\mathbf{m_f}, \mathbf{S_f}\xi)$, where $\mathbf{m_f}$ and $\mathbf{S_f}$ are variational parameters, and $q(\xi; \boldsymbol{\varphi_\xi})$ and $q(\omega; \boldsymbol{\varphi_\omega})$ are parameterized with any positive distribution such as a normalizing flow. We use this approximation when we derive the predictive distribution

$$p(f^*|\mathbf{y}) = \int p(f_*|\boldsymbol{f}, \xi; \boldsymbol{\eta})p(\boldsymbol{f}, \xi|\mathbf{y}; \boldsymbol{\eta})d\boldsymbol{f}d\xi \tag{52}$$

$$= \int p(f_*|\boldsymbol{f}, \xi; \boldsymbol{\eta_f})p(\boldsymbol{f}, \xi|\mathbf{y}; \boldsymbol{\eta})d\boldsymbol{f}d\xi \tag{53}$$

$$\approx \int p(f_*|\boldsymbol{f}, \xi; \boldsymbol{\eta_f})q(\boldsymbol{f}|\xi; \boldsymbol{\varphi_f})q(\xi; \boldsymbol{\varphi_\xi})d\boldsymbol{f}d\xi. \tag{54}$$

$$\tag{55}$$

If we first take a look at the prior distribution $p(f^*, \boldsymbol{f}|\xi)$ when $\xi$ is constant, which is

$$\begin{bmatrix} f^* \\ \boldsymbol{f} \end{bmatrix} \xi \sim \mathcal{N} \left( 0, \begin{bmatrix} k_{**} & \mathbf{k}_*^\top \\ \mathbf{k}_* & \mathbf{K} \end{bmatrix} \xi \right), \tag{56}$$

with the the conditional distribution

$$p(f^*|\boldsymbol{f}, \xi; \boldsymbol{\eta}) = \mathcal{N} \left( \mathbf{k}_*^\top \mathbf{K}^{-1} \boldsymbol{f}, \left( k_{**} - \mathbf{k}_*^\top \mathbf{K}^{-1} \mathbf{k}_* \right) \xi \right) \tag{57}$$

$$= \mathcal{N} \left( \mathbf{a}^\top \boldsymbol{f}, b\xi \right). \tag{58}$$

Here, $\mathbf{a}^\top = \mathbf{k}_*^\top \mathbf{K}^{-1}$ and $b = k_{**} - \mathbf{k}_*^\top \mathbf{K}^{-1} \mathbf{k}_*$. We use this expression and the variational approximation when we derive the posterior predictive distribution,

$$p(f^*|\mathbf{y}) = \int p(f_*|\boldsymbol{f}, \xi; \boldsymbol{\eta}) q(\boldsymbol{f}|\xi; \varphi_{\boldsymbol{f}}) q(\xi; \varphi_\xi) d\boldsymbol{f} d\xi \tag{59}$$

$$= \mathbb{E}_{q(\xi; \boldsymbol{\varphi}_\xi)} \left[ \int p(f_*|\boldsymbol{f}, \xi) q(\boldsymbol{f}|\xi; \varphi_{\boldsymbol{f}}) d\boldsymbol{f} \right] \tag{60}$$

$$= \mathbb{E}_{q(\xi; \boldsymbol{\varphi}_\xi)} \left[ \int \mathcal{N} \left( f_*|\mathbf{a}^\top \boldsymbol{f}, b\xi \right) \mathcal{N} \left( \boldsymbol{f}|\mathbf{m}, \mathbf{S}\xi \right) d\boldsymbol{f} \right] \tag{61}$$

$$= \mathbb{E}_{q(\xi; \boldsymbol{\varphi}_\xi)} \left[ \int \mathcal{N} \left( f_*|\mathbf{a}^\top \mathbf{m}, \mathbf{a}^\top \mathbf{S} \mathbf{a}\xi + b\xi \right) \right] \tag{62}$$

$$= \mathbb{E}_{q(\xi; \boldsymbol{\varphi}_\xi)} \left[ \mathcal{N}(f_*|m_*, s_*\xi) \right] \tag{63}$$

where

$$m_* = \mathbf{a}^\top \mathbf{m} \tag{64}$$

$$s_* = \mathbf{a}^\top \mathbf{S} \mathbf{a} + b \tag{65}$$

and we get the covariance by $\mathbb{E}[\xi] s_*$.

**Optimizing the ELBO**

We train the model by optimizing the evidence lower bound (ELBO) given by

$$\mathcal{L}(\boldsymbol{\varphi}, \boldsymbol{\eta}) = \mathbb{E}_{q(\boldsymbol{f}|\xi; \varphi_{\boldsymbol{f}})q(\xi; \varphi_\xi)q(\omega; \varphi_\omega)} \left[ \log p(\mathbf{y}, \boldsymbol{f}, \xi, \omega; \boldsymbol{\eta}) - \log \left( q(\boldsymbol{f}|\xi; \varphi_{\boldsymbol{f}}) q(\xi; \varphi_\xi) q(\omega; \varphi_\omega) \right) \right]. \tag{66}$$

The model is implemented in Pyro (Bingham et al., 2018), see Section F for details.

# E   Sparse elliptical processes

With the variational inference framework we can create a sparse version of the model

$$\int p(\boldsymbol{f}, \boldsymbol{u}, \xi; \boldsymbol{\eta}) d\xi = \int p(\boldsymbol{f}|\boldsymbol{u}, \xi; \boldsymbol{\eta_f}) p(\boldsymbol{u}|\xi; \boldsymbol{\eta_u}) p(\xi; \boldsymbol{\eta_\xi}) d\xi, \tag{67}$$

where $\boldsymbol{u}$ are outputs of the elliptical process, located at the inducing inputs $\mathbf{x}_u$. We approximate the posterior with

$$p(\boldsymbol{f}, \boldsymbol{u}, \xi|\mathbf{y}; \boldsymbol{\eta}) \approx p(\boldsymbol{f}|\boldsymbol{u}, \xi; \boldsymbol{\eta_f}) q(\boldsymbol{u}|\xi; \boldsymbol{\varphi_u}) q(\xi; \boldsymbol{\varphi_\xi}) \tag{68}$$

The posterior of the distribution is given by

$$p(f^*|\mathbf{y}) = \int p(f_*|\boldsymbol{f}, \boldsymbol{u}, \xi; \boldsymbol{\eta}) p(\boldsymbol{f}, \boldsymbol{u}, \xi|\mathbf{y}; \boldsymbol{\eta}) d\boldsymbol{f} d\boldsymbol{u} d\xi$$

$$\approx \int p(f_*|\boldsymbol{f}, \boldsymbol{u}, \xi; \boldsymbol{\eta}) p(\boldsymbol{f}|\boldsymbol{u}, \xi; \boldsymbol{\eta_f}) q(\boldsymbol{u}|\xi; \boldsymbol{\varphi_u}) q(\xi; \boldsymbol{\varphi_\xi}) d\boldsymbol{f} d\boldsymbol{u} d\xi$$

$$= \int \left[ \int p(f_*|\boldsymbol{f}, \boldsymbol{u}, \xi; \boldsymbol{\eta}) p(\boldsymbol{f}|\boldsymbol{u}, \xi; \boldsymbol{\eta_f}) d\boldsymbol{f} \right] q(\boldsymbol{u}|\xi; \boldsymbol{\varphi_u}) q(\xi; \boldsymbol{\varphi_\xi}) d\boldsymbol{u} d\xi \tag{69}$$

We can simplify the inner expression by using the fact that the elliptical distribution is consistent,

$$\int p(f_*|\boldsymbol{f}, \boldsymbol{u}, \xi; \boldsymbol{\eta}) p(\boldsymbol{f}|\boldsymbol{u}, \xi; \boldsymbol{\eta}) d\boldsymbol{f} = \int p(f_*, \boldsymbol{f}|\boldsymbol{u}, \xi; \boldsymbol{\eta}) d\boldsymbol{f} = p(f_*|\boldsymbol{u}, \xi; \boldsymbol{\eta}). \tag{70}$$

Hence, Equation (69) is simplifies to

$$p(f^*|\mathbf{y}) = \int p(f_*|\boldsymbol{u}, \xi; \boldsymbol{\eta}) q(\boldsymbol{u}|\xi; \boldsymbol{\varphi_u}) q(\xi; \boldsymbol{\varphi_\xi}) d\boldsymbol{u} d\xi, \tag{71}$$

where $q(\boldsymbol{u}|\xi; \boldsymbol{\varphi_u}) = \mathcal{N}(\mathbf{m}_u, \mathbf{S}_u \xi)$ with the variational parameters $\mathbf{m}_u$ and $\mathbf{S}_u$, and $\xi$ is parameterized, e.g., by a normalizing flow

Finally, we obtain the posterior $p(f^*|\boldsymbol{x}^*) = \mathbb{E}_{q(\xi; \boldsymbol{\varphi_\xi})} \left[ \mathcal{N}(f_n|\mu_{\boldsymbol{f}}(\boldsymbol{x}^*), \sigma_{\boldsymbol{f}}(\boldsymbol{x}^*)) \right]$ where

$$\mu_{\boldsymbol{f}}(\boldsymbol{x}_n) = \boldsymbol{k}_n^\top \boldsymbol{K_{uu}}^{-1} \boldsymbol{m} \tag{72}$$

$$\sigma_{\boldsymbol{f}}(\boldsymbol{x}_n) = k_{nn} - \boldsymbol{k}_n^\top \left( \boldsymbol{K_{uu}}^{-1} - \boldsymbol{K_{uu}}^{-1} \boldsymbol{S} \boldsymbol{K_{uu}}^{-1} \right) \boldsymbol{k}_n. \tag{73}$$

Here $\boldsymbol{k}_n = k(\boldsymbol{x}_n, \boldsymbol{X_u})$, $k_{nn} = k(\boldsymbol{x}_n, \boldsymbol{x}_n)$, and $\boldsymbol{K_{uu}} = k(\boldsymbol{X_u}, \boldsymbol{X_u})$.

## F  Implementation: variational inference

We used the Pyro library (Bingham et al., 2018), which is a universal probabilistic programming language (PPL) written in Python and supported by PyTorch on the backend.

In Pyro, we trained a model with variational inference (Kingma & Welling, 2013) by creating "stochastic functions" called **model** and a **guide**, where the **model** samples from the prior latent distributions $p(\boldsymbol{f}, \xi, \omega; \boldsymbol{\eta})$, and the observed distribution $p(\mathbf{y}|\boldsymbol{f}, \omega)$, and the **guide** samples the approximate posterior $q(\boldsymbol{f}|\xi; \boldsymbol{\varphi_f}) q(\xi; \boldsymbol{\varphi_\xi}) q(\omega; \varphi_\omega)$. We then trained the model by minimizing the ELBO, where we simultaneously optimized the model parameters $\boldsymbol{\eta}$ and the variational parameters $\boldsymbol{\varphi}$. (See more details here, `https://pyro.ai/examples/svi_part_i.html`.)

To implemented the model in Pyro, we created the guide and the model (see Algorithm 3), which we did by building upon the already implemented variational Gaussian process. We used the guide and the model to derive the evidence lower bound (ELBO), which we then optimized with stochastic gradient descent using the Adam optimizer (Kingma & Ba, 2015).

We used the already implemented rational linear spline flow for the normalizing flow in Pyro.

---

**Algorithm 1** PyTorch implementation of the variational sparse elliptical process (VI-$\mathcal{EP}$-$\mathcal{EP}$).

---

1: **procedure** MODEL(**X**, **y**)
2:     Sample $\xi = $ from $p(\xi; \boldsymbol{\eta}_\xi)$( Normalizing flow )
3:     Sample $\boldsymbol{u}$ from $\mathcal{N}(\boldsymbol{0}, \xi \boldsymbol{K_{uu}}))$                      ▷ Take a sample from the latent $\boldsymbol{u}, \xi$ and $\omega$
4:     Derive the variational posterior $\prod_{n=1}^{N} q(f_n|\xi; \boldsymbol{\varphi}) = \mathcal{N}(\mu_{\boldsymbol{f}}(\boldsymbol{x}_n), \sigma_{\boldsymbol{f}}(\boldsymbol{x}_n)\xi)$.    ▷ During training $\xi$ is sampled from
    the posterior/guide.
5:     Take a Monte-Carlo sample $\hat{f}_n$ from each $q(f_n|\xi; \varphi)$
6:     Sample for each $\boldsymbol{x}_n \zeta_n = $ from $\mathcal{N}(0, 1)$
7:     Derive $\omega_n = T(\zeta_n; \boldsymbol{\eta}_\omega)$
8:     Derive the log probability of $\prod_{n=1}^{N} \mathcal{N}(y_n|\hat{f}_n, \omega_n)$        ▷ during training $\omega_n$ is sampled from the posterior/guide.
9: **end procedure**
10: **procedure** GUIDE
11:     Sample $\xi = $ from $q(\xi; \boldsymbol{\varphi}_\xi)$( Normalizing flow )
12:     Sample $\boldsymbol{u}$, from $\mathcal{N}(\mathbf{m}, \mathbf{S}\xi))$
13:     For each $\boldsymbol{x}_n$ sample $\zeta$ from $\mathcal{N}(\mu_\zeta((y_n - f_n)^2), \sigma_\zeta((y_n - f_n)^)).$
14: **end procedure**

---

---

**Algorithm 2** PyTorch implementation of the variational sparse parametric elliptical process (PP-$\mathcal{EP}$-$\mathcal{EP}$).

---

1: **procedure** MODEL($\mathbf{X}, \mathbf{y}$)
2:     Sample $\xi =$ from $p(\xi; \boldsymbol{\eta}_\xi)$( Normalizing flow )
3:     Sample $\boldsymbol{u}$ from $\mathcal{N}(\mathbf{0}, \xi \boldsymbol{K_{uu}})$)                    ▷ Take a sample from the latent $\boldsymbol{u}, \xi$ and $\omega$
4:     Derive the variational posterior $\prod_{n=1}^{N} q(f_n|\xi; \boldsymbol{\varphi}) = \mathcal{N}(\mu_{\boldsymbol{f}}(\boldsymbol{x}_n), \sigma_{\boldsymbol{f}}(\boldsymbol{x}_n)\xi)$.    ▷ During training $\xi$ is sampled from
     the posterior/guide.
5:     For each $\boldsymbol{x}_n$ sample $\zeta_n =$ from $\mathcal{N}(0, 1)$
6:     Derive $\omega_n = T(\zeta_n; \boldsymbol{\eta}_\omega)$
7:     Derive the log probability of $\prod_{n=1}^{N} \mathcal{N}(y_n|\mu_{\boldsymbol{f}}(\boldsymbol{x}_n), \sigma_{\boldsymbol{f}}(\boldsymbol{x}_n)\xi + \omega_n)$        ▷ during training $\omega_n$ is sampled from the
     posterior/guide.
8: **end procedure**
9: **procedure** GUIDE
10:     Sample $\xi =$ from $q(\xi; \boldsymbol{\varphi}_\xi)$( Normalizing flow )
11:     Sample $\boldsymbol{u},$ from $\mathcal{N}(\mathbf{m}, \mathbf{S}\xi)$)
12:     For each $\boldsymbol{x}_n$ sample $\zeta$ from $\mathcal{N}(\mu_\zeta(\boldsymbol{z}), \sigma_\zeta(\boldsymbol{z}))$. Were $\boldsymbol{z} = [(y_n - \mu_{\boldsymbol{f}}(\boldsymbol{x}_n))^2), \sigma_{\boldsymbol{f}}(\boldsymbol{x}_n)]$.
13: **end procedure**

---

## G   Regression experiment setup

In the regression experiments in section 4.2 we ran all experiments using the Adam optimizer (Kingma & Ba, 2015) with a learning rate of 0.003 that was sequentially decreased during the training. For all experiments, we created ten random train/val/test splits with the proportions 0.75/0.1/0.15, except for the two smallest datasets (machine and mpg), where we instead evaluated the model on the training data (the split proportions was train/test =0.75/0.25). We used the model with the lowest predictive probability on the validation set. For the large datasets ($n > 1000$), we used 500 inducing points and a batch size of 1000. For the small datasets, we used 100 inducing points and no batching. We run the training for the large dataset during 250 epochs and the small dataset for 5000 epochs. For the full GP, we used a learning rate of 0.01, which we decreased during the training. For the large datasets (n>1000), we trained the full $\mathcal{GP}$ on a single split.

**Elliptical process setup.**   The likelihood mixing distribution uses a spline flow with 9 bins and *Softplus* as its output transformation. The elliptic posterior mixing distribution uses a spline flow with 5 bins and a *Sigmoid* otput transformation. The reason we use a *Sigmoid* for the process is that we want to regularize it more since we hypothesis it is more difficult to learn. The neural network of the posterior likelihood mixing distribution uses a two layer neural network with 32 hidden dimensions.

## H   Results

The regression results from figures 7 are presented in tables 2 and 3.

Table 2: Predictive Mean Square Error (MSE) on the hold out sets from the experiments. We show the average of the ten runs and one standard deviation in parenthesis.

|          | Machine        | MPG            | Concrete       | Elevators      | California     | Kin40k         | Protein        |
|----------|----------------|----------------|----------------|----------------|----------------|----------------|----------------|
| PP-EP-EP | 0.472 (0.516)  | 0.111 (0.029)  | 0.131 (0.025)  | 0.166 (0.006)  | 0.230 (0.013)  | 0.106 (0.004)  | 0.506 (0.007)  |
| PP-EP-GP | 0.503 (0.478)  | 0.121 (0.027)  | 0.123 (0.009)  | 0.170 (0.006)  | 0.237 (0.013)  | 0.139 (0.003)  | 0.533 (0.012)  |
| PP-GP-GP | 0.395 (0.304)  | 0.125 (0.019)  | 0.154 (0.015)  | 0.173 (0.006)  | 0.249 (0.013)  | 0.176 (0.006)  | 0.575 (0.006)  |
| VI-EP-EP | 0.580 (0.518)  | 0.108 (0.022)  | 0.134 (0.024)  | 0.155 (0.005)  | 0.255 (0.015)  | 0.154 (0.003)  | 0.651 (0.009)  |
| VI-EP-GP | 0.274 (0.460)  | 0.117 (0.025)  | 0.122 (0.013)  | 0.160 (0.005)  | 0.261 (0.015)  | 0.152 (0.004)  | 0.647 (0.013)  |
| VI-GP-GP | 0.261 (0.219)  | 0.121 (0.027)  | 0.119 (0.015)  | 0.157 (0.005)  | 0.251 (0.011)  | 0.170 (0.003)  | 0.652 (0.013)  |
| Exact GP | 0.417 (0.417)  | 0.123 (0.123)  | 0.098 (0.098)  | 0.175 (0.175)  | 0.227 (0.227)  |                |                |

Table 3: Negative log likelihood on the hold out sets from the experiments. We show the average of the ten runs and one standard deviation in parenthesis.

|          | Machine        | MPG           | Concrete      | Elevators     | California    | Kin40k        | Protein       |
|----------|----------------|---------------|---------------|---------------|---------------|---------------|---------------|
| PP-EP-EP | -0.053 (0.215) | 0.168 (0.076) | 0.247 (0.075) | 0.393 (0.012) | 0.453 (0.015) | 0.247 (0.010) | 0.975 (0.007) |
| PP-EP-GP | -0.206 (0.315) | 0.215 (0.087) | 0.180 (0.062) | 0.397 (0.011) | 0.464 (0.016) | 0.312 (0.005) | 1.006 (0.010) |
| PP-GP-GP | -0.403 (0.245) | 0.220 (0.066) | 0.359 (0.050) | 0.426 (0.009) | 0.554 (0.017) | 0.387 (0.007) | 1.060 (0.004) |
| VI-EP-EP | 0.237 (0.559)  | 0.240 (0.082) | 0.379 (0.085) | 0.466 (0.011) | 0.618 (0.020) | 0.527 (0.010) | 1.209 (0.006) |
| VI-EP-GP | -0.134 (0.250) | 0.265 (0.066) | 0.340 (0.042) | 0.481 (0.005) | 0.633 (0.038) | 0.558 (0.009) | 1.207 (0.009) |
| VI-GP-GP | -0.063 (0.259) | 0.339 (0.091) | 0.339 (0.052) | 0.491 (0.010) | 0.730 (0.018) | 0.608 (0.005) | 1.210 (0.009) |
| Exact GP | -0.135 (-0.135)| 0.438 (0.438) | 0.180 (0.180) | 0.520 (0.520) | 0.771 (0.771) |               |               |

## I  Datasets

**Elevators dataset**   (Dua & Graff, 2017) is obtained from the task of controlling a F16 aircraft, and the objective is related to an action taken on the elevators of the aircraft according to the status attributes of the aeroplane.

**Physicochemical properties of protein tertiary structure dataset**   The data set is taken from CASP 5-9. There are 45730 decoys and size varying from 0 to 21 armstrong.

**California housing dataset**   was originally published by Pace & Barry (1997). There are 20 640 samples and 9 feature variables in this dataset. The targets are prices on houses in the California area.

**The Concrete dataset**   (Yeh, 1998) has 8 input variables and 1030 observations. The target variables are the concrete compressive strength.

**Machine CPU dataset**   (Kibler et al., 1989) where the target value is the relative performance of the CPU. The dataset consist of 209 samples with nine attributes.

**Auto MPG dataset** (Alcalá-Fdez et al., 2011) originally from the StatLib library which is maintained at Carnegie Mellon University. The data concerns city-cycle fuel consumption in miles per gallon and consists of 392 samples with five features each.

**Pima Indians Diabetes Database**   (Smith et al., 1988) originally from the National Institute of Diabetes and Digestive and Kidney Diseases. The objective of the dataset is to diagnostically predict whether or not a patient has diabetes, based on certain diagnostic measurements included in the dataset. The dataset consist of 768 samples with eight attributes.

**The Cleveland Heart Disease dataset**   consists of 13 input variables and 270 samples. The target classifies whether a person is suffering from heart disease or not.

**The Mammographic Mass dataset**   predicts the severity (benign or malignant) of a mammographic mass lesion from BI-RADS attributes and the patient's age. This dataset consists of 961 with six attributes.

