# OpenReview forum: "Variational Elliptical Processes"
_TMLR — Rejected by TMLR_

### Review · Reviewer_9XiL · 2022-06-07

**Summary Of Contributions:**

This paper claims to:
- construct elliptical processes as a distribution on functions
- heteroskedastic likelihood functions based on eliptically distributed noise
- inducing point based variational inference

The novelty in the construction of elliptical processes is the flexible parameterisation of the distribution on the scale parameter $\xi$.

**Broader Impact Concerns:**

None.

**Requested Changes:**

## Citations
There are a few strange citations, which I would suggest that they should be changed:
- Page 1: Tran et al 2016. The proposed variational elliptical process method uses approximate posteriors in a way almost exactly the same as in regular approximate GPs (e.g. Hensman et al 2015, which is cited, and Hensman et al 2013, GPs for big data, which isn't cited). This is a very different setting to the Tran et al 2016 paper, which aims somehow to use a GP in a variational distribution to parameterise more complex distributions. I think this citation is not appropriate here.
- Page 1 and elsewhere: Student's t processes were discussed significantly before Shah et al 2014. See §9.9 in _Gaussian Processes for Machine Learning_ by Rasmussen & Williams. A more comprehensive list of citations should be included.
- Page 6: It's mentioned that BBVI is used, and Wingate & Weber, Ranganath et al are cited. However, in the appendix it is stated that Pyro is used, and references are given to the reparameterisation trick, which is not used in the W&W and Retal papers. These papers can be mentioned in related work on BBVI, but I would suggest for the method that is _actually_ used to be cited in the main explanation of the method.
- There is one unpublished preprint on elliptical processes which is not cited: https://arxiv.org/abs/2003.00720. Since this is very relevant, I do think it should be cited.

## Experiments
I believe that the experiments need to be significantly expanded, in accordance with what is mentioned above. Particularly the comparison to a full GP is important, because a) models should not be artificially limited in capacity when benchmarking, and b) so the effect of the change in prior can be investigated independently of effects from the approximation.

**Strengths And Weaknesses:**

The paper is clearly written, with only a few passages where it would be nice to have more details.

While elliptical processes have been studied before (see unpublished preprint https://arxiv.org/abs/2003.00720), the paper does contain some novelties. In particular, the flexible distributions on scale parameters, and variational approximations.

For this paper to be useful, the experiments need to really clearly show the situations in which this methodology is useful. This is where the paper is currently lacking, and I believe this needs to be improved before the paper can be useful to the ML community.

§4.1 contains a useful synthetic experiment on showing that heavy-tailed noise can be dealt with well. This could be improved by including an example of how a GP with the typical Gaussian noise behaves poorly.

§4.2 contains some real datasets, on which EPs are illustrated. The datasets are small, although this is not a bad thing per se. Working with small datasets is a sensible research question. The main issue is that the experiments are superficial, in that they only provide performance metrics, without providing clear ablations of the different components of the proposed method. I can see that there was some attempt to do this, based on the two EP models that were introduced. However, it is not clear what questions these comparisons are trying to address. In addition it seems strange that EP^2 uses both a GP prior and posterior. Why call this an EP? Is this a typo?

This section really should be a key support of the paper. To improve this section, I suggest addressing the following questions:
1. Does an EP prior on the latent function improve performance, as compared to a GP? The comparison really should be to a _full GP_, without any inducing point approximations, to ensure that _model_ assumptions are tested, not any error that is introduced by the approximation. All datasets (even the 20k one) can be run using a full Cholesky decomposition on modern desktop machines. Using 20 or 50 inducing points is not enough.
2. Does an elliptical noise distribution help with prediction? For this you can compare a GP with Gaussian noise to a GP with elliptical noise.
3. How much does both an EP and elliptical noise further improve performance?

Extending the range of datasets to include more of the UCI datasets will also be helpful.

Similar issues surrounding a lack of inducing points also hold in the classification settings of §4.3. In addition, it is rather unusual to provide the result for each fold in a run of crossvalidation, as is done in fig 5. Would it not be better to present the results in a table, with the averaged results being provided? It is perhaps likely that the differences won't look very large after averaging. Perhaps this is not very surprising in classification, as an EP effectively only performs Bayesian inference over the scale parameter, which classification isn't very sensitive too. It is sensitive in cases where data is scarce and there are somewhat overlapping classes, which could be the case in the datasets that are investigated.

---

> ### Author Response · Authors · 2022-07-03
> **Response after revision**
>
> **Experiments**
> >§4.1 contains a useful synthetic experiment on showing that heavy-tailed noise can be dealt with well. This could be improved by including an example of how a GP with the typical Gaussian noise behaves poorly.
>
> Thank you for the suggestion, we have added figure 6 to show this.
>
> We are also grateful for your excellent suggestions on how to improve §4.2. As requested, we have included additional (larger) datasets and added comparisons of different variations of the EP with an exact GP. The results show that EPs (with 500 inducing points) do indeed perform better, especially on the larger datasets.
>
> In §4.3 we’ve updated the presentation of the results. We hope you find it more appropriate. On the other hand, these dataset are too small to increase the number of inducing points significantly.
>
> **Citations**
>
> In the introduction, we removed Tran (2016) and added a reference to Rasmussen & Williams (2006). Shah et al. lists a number of works that used Student-t processes before them (Yu et al., 2007, Zhang and Yeung, 2010, Xu et al., 2011, Archambeau and Bach, 2010) but since we consider these only vaguely related we prefer to leave them out. Student-t processes have also been discussed much earlier in the Bayesian statistics literature, and we’ve supplemented the references in the related work section with one to O’Hagan (1991). We also included references to Hensman et al. (2013) and to the Pyro library (Bingham et al, 2019).

---

### Review · Reviewer_CEkx · 2022-06-20

**Summary Of Contributions:**

The paper proposes *elliptical processes*, a new family of probabilistic stochastic process on top of Gaussian processes (GPs) and Student-T processes (SPs), whose underlying mechanism is based on the elliptical distribution. For performing inference over such processes, the paper develops approximate methods and uses modern density estimators; in particular variational inference and normalizing flows with spline mappings. To achieve both inference and predictive tasks, the methodology faces several issues concerning the tractability of integrals, which in the end are circumvent via approximations. The paper also proposes two extensions for sparse approximations (to scale up the model) and *heteroscedastic* noise. Experimental results show evidence of performance in regression, classification and *heteroscedastic* tasks.

**Broader Impact Concerns:**

I do not detect particular ethical implications of the work. A slight detail would be around this claim that heavier tailed posteriors are of interest for some applications. This is perhaps odd, as we generally want systems to be more certain on the tasks for safety reasons. I think it would be helpful to have some comments on the aspect between heavier tails and uncertainty.

**Requested Changes:**

To me, the manuscript would have to be improved on three main points:

**Change 1**: Introduce, motivate and explain better why heavier tails are desired and why the reader should be interested on them. Connect that point with uncertainty and which properties one desires to achieve when having both an elliptical process with an elliptical likelihood model on top. Perhaps, it is too much, but if the EP is a general extension of GPs and SPs, would be nice to have more references to this connection and what does it imply.

**Change 2**: Section 3 should be re-written to be clearer and cleaner on the message. At least to make it clear why decisions are taken and why the likelihood is on the likelihood noise. And why, for instance, we observe this noise (I think authors mention something like this). At its current state, this is not clear enough.

**Change 3**: I would suggest to improve a bit the experiments, at least with toy experiments where it is very clear that the EP is doing what authors claim (to capture an underlying process with heavier tails). Also, I would add more details and results on the computational cost and perhaps on the sparse extension that is mentioned. The classification results on AUC are a bit odd for the GP community, so extra metrics or visual results would also help. In general, there is quite a lot of gap in the experiments for improvement, which I consider would be good for the manuscript.

**Strengths And Weaknesses:**


**Strengths**
> In my opinion, the paper tackles an interesting idea and is in general on a good direction. Authors seems to know well the literature, existing approaches and the spirit is somehow similar to Shah (ICML 2014). This last point is also positive to me. The construction of the whole approach based in the elliptical distribution seems technically correct and I did not miss any particular detail in the first part of the manuscript. Despite some little inconsistencies on the notation, the formulation is somehow clear and well introduced. The decisions taken for inference and the methods used are meaningful and similar to the strategies taken in the state-of-the-art, which makes it reliable. In the experiments, authors positively address three different tasks to characterise the performance of the method.

**Weaknesses**
> Even considering that the paper is well-written and thorough in many aspects, there is a general doubt that comes to mind when reading the paper. In the original Student T paper of Shah (ICML 2014), the idea was to explore other alternatives to circumvent the limitation of the *Gaussianity* (a lot of probability mass around the mean, but tails quickly going to zero when euclidean distance augments). The idea of considering Student T processes was interesting back then because one could obtain posterior densities with tighter or heavier tails depending on which type of process one wants. In the end, this translated to more certainty, or at least that was my impression. On the other hand, the pursue of heteroscedastic GPs is also of interest in the community, due to one might want different noise levels depending on the input region. However, I detect a general satisfaction in the paper with having heavier and heavier tails, which in the end, looks like having more uncertain in the model to me. I say this, because I do not fully understand the motivation behind this search, or at least I would like to comprehend. This might come from a bit of lack of motivation on the application side, e.g. what is the specific scenario where a heavy tail process wins the GP. Another example of this can be found in the second paragraph of the Conclusion, where authors indicate

```
"The variational approximation we propose enables us to capture heavy-tailed posteriors and makes it straightforward to create a sparse variational elliptical process that scales to large datasets"
```
This is a bit odd to me, since one generally desires a posterior which is more certain, with the less probability around its mean as possible. Claiming that a posterior distribution with heavier tails seems to go into the opposite direction.

Other aspect that concerns me is the presentation of the methodology in **Section 3**. I think this could be improved significantly, at least to make it clearer why the decisions are taken. In particular, I do not understand the modelling of the likelihood noise $\epsilon_n$ in the likelihood subsection instead of the output data points $y_n$. Later, the normalising flow is introduced here, what makes things a bit more difficult to get.

The last point of weakness, to me, is that I find experimental results somehow limited. In particular, the toy regression task seems to be too simple with the sinusoidal function and only N=300 datapoints. Also, I would love to find a clear result that motivates the search of heavier tails in the stochastic process, but somehow I do not see it. Additionally, the $\mathcal{EP}^2$ model used in the following regression experiments looks unclear to me, as it is indicated to use both a GP-prior and a GP-posterior. Why is a $\mathcal{EP}$ then? In general, even if the experiments look on a good direction, addressing three tasks, they do not project an idea about the computational cost or for example the difficulty to fit the model. This could be improved, quite a lot.

---

> ### Author Response · Authors · 2022-07-03
> **Response after revision**
>
> Thanks for your comments and suggestions. Regarding the request changes:
>
> **Change 1**:
> We certainly want the predictive uncertainty to be small _given that it corresponds to the true uncertainty_. We have extended the first paragraph of the introduction with the argument that accurate uncertainty quantification is important for autonomous decision-making. In the experiments, we have also added figure 6 in attempt to illustrative how a GP fitted to heavy-tailed (specifically, elliptic) noise can be overconfident in its predictions. This is also highly related to the work of Jankowiak et al., where they argue that standard variational inference can lead to similar problems.
>
> The elliptical process prior is mainly relevant when encountering multi-path data, for which we have added a method and experiments.
>
> **Change 2**:
> We have made a major overhaul of section 3 in an attempt to make it clearer and cleaner. We hope you find it satisfactory.
>
> **Change 3**:
> We have extended in particular the regression experiments quite substantially, and changed the presentation format of the classification results as well as including also the accuracy.

---

> > ### Comment · Reviewer_CEkx · 2022-07-21
> > **After revision comments**
> >
> > Thanks to the authors for the effort put into the revision of the manuscript. I have to acknowledge that the paper has significantly improved and is now closer to the quality threshold for acceptance. Particularly, I am satisfied with the changes introduced in **Section 3**, where is now clearer what are the technical contributions and more importantly, with the additional results in the **Experiments**. I would have loved to see a larger motivation for elliptical processes and why exactly authors desire this type of probabilistic stochastic process, but I also accept that authors want it to be like this.
> >
> > I partially agree with the *Reviewer 9XiL* in what is said about the experimental results. Even being a sufficient amount to prove the performance of the method, they could be improved to answer the key questions that introducing EPs implies.
> >
> > Anyway, if these details on the experiments could be improved now or in the future, I would clearly vote for clear accept.

---

### Review · Reviewer_fi2A · 2022-06-21

**Summary Of Contributions:**

The purpose of the paper is to demonstrate efficient large-scale modeling and inference with elliptical distributions and elliptical processes. The main modeling insight is that a consistent elliptic distribution/process is a scale mixture of Gaussians; so, the conditional distributions are Gaussian (conditioned on the mixing variable, that is). This insight is leveraged by parameterizing the mixing distribution as a normalizing flow and using variational inference over the resulting family to form approximate posteriors. Large-scale inference is accomplished by inducing variables (as done in sparse Gaussian processes). A heteroscedastic noise model with input-dependent normalizing flow parameter(s) is also described. Regression and classification results on synthetic and real datasets are provided.

**Broader Impact Concerns:**

No impact statement was provided, but I don't see any broader impact concerns with this work.

**Requested Changes:**

Critical:

* Please include some baselines in the experimental section. There are many to choose from. Also the paper needs to include similar experiments on large datasets. For example, "Parametric Gaussian Process Regressors" (2020) by Jonkowiak et al. lists numerous medium-to-large datasets that could be utilized. The synthetic heteroscedastic experiment was cool, but perhaps the proposed method could be compared e.g., to Table 2 in "Variational Heteroscedastic Gaussian Process Regression"? Also, why not compare how variational elliptical processes due in robust regression like in "Robust Gaussian Process Regression with a Student-t Likelihood"? To be clear, I'm not suggesting running all the baselines above; I'm suggesting running the proposed method in the same experimental settings as those baselines and comparing against published results.

* Please provide some discussion of practical usability of the proposed method. How does one select the normalizing flow parameterization for a novel dataset? Training time comparisons to baselines would be great to include in the supplement or experiments.

Non-critical:

* Could the authors clear up the confusing points? One was a brief section on ergodicity. I believe the paper was just stating that the mixing distribution cannot be learned from data that represents a single draw of an elliptical process. However, the section seemed to imply that either mixture processes or non-stationary processes (or both?) cannot be estimated which would be helped by an illustrative example or two. Gaussian processes with non-stationary kernels are routinely used and estimation in a mixture model is essentially Type 2 ML. What is the effect of this point of ergodicity on the experimental conclusions? Perhaps I am missing something obvious here? The second confusing point was the statement on the first half of page 4: "The conditional distribution is guaranteed to be a consistent elliptical distribution but not necessarily the same as the original one..." How does this issue specifically manifest in inference? Could an example be provided?

* Describe how the predictive mean (RHS of (16), (20)) is calculated for the experiments.

**Strengths And Weaknesses:**

Strengths:

* Since the elliptical distribution subsumes many other thin- and heavy-tailed distributions, efficient and algorithmically robust/stable inference in this family is a very attractive proposition. The conditionalization property highlighted is nicely leveraged by the proposed variational inference approach. That it also leads directly to a sparse formulation is kind of neat too! The paper hints at a strong foundation here.

* The paper's exposition was fairly clear: laid out well and mostly easy to follow.

Weaknesses:

* The biggest weakness is lack of compelling evidence to use the proposed model and inference approach. First, the reported results in Table 1 do not show a strong statistically significant performance advantage of the proposed method over a standard GP (wrt/ reported error bars). The classification results are reported in a per-fold manner and show maybe a slight advantage for the proposed method, but not significant. Second, there are no competitors included in the experimental results, e.g., sparse variational GPs, full GPs with variational approximation, etc. If the heavy-tail modeling capability of elliptic distributions was to be spotlighted, an application in robust regression with real data would have been expected somewhere. Third, there are virtually no experiments on medium-to-large datasets. With the exception of the California dataset, all of the datasets are small (1030 sample size or less). A sparse approximation, as utilized in the experimental section, should not be required in these cases (unless I am mis-understanding something about the proposed model and inference approach).

* A second weakness (less substantial than the previous) is lack of any discussion on how difficult or easy hyperparameter selection with variational elliptical processes is. For example, a standard issue with sparse variational GPs with non-Gaussian likelihoods is how finicky hyperparameter optimization vs. variational parameter optimization can be. Another issue that a reader may be curious about is how stable variational EM is in this scenario when modeling the mixing distribution with a normalizing flow.

* There were two confusing points in the exposition regarding ergodicity and the conditional distribution that hindered understanding a little.

---

> ### Author Response · Authors · 2022-07-04
> **Response after revision**
>
> Thanks for your appreciative comments and helpful suggestions on improvements!
>
> > Please include some baselines in the experimental section.
>
> We have added comparisons with the Parametric GP (Jankowiak et al) and an exact GP, and also tried to clarify that we also compare with SVGP.
>
> > the paper needs to include similar experiments on large datasets. For example, "Parametric Gaussian Process Regressors" (2020) by Jonkowiak et al. lists numerous medium-to-large datasets that could be utilized.
>
> Thanks for the reference! We have added results on several of those datasets, and also developed and evaluated an adaptation of their training method for EPs. The results show that the EP-models generally result in higher predictive likelihoods, especially on the (new) larger datasets.
>
> > Please provide some discussion of practical usability of the proposed method.
>
> We have added more experimental details in the appendix and plan to extend this further in the final version (including training times). Since our primary goal hasn’t necessarily been to achieve state-of-the-art results, we’ve only put a modest effort into the hyperparameter tuning. For this reason, we did not include a discussion on it, but most likely it would lead to slight gains in performance.
>
> > How does one select the normalizing flow parameterization for a novel dataset?
>
> We train the flow and model parameters together using stochastic gradient descent.
>
> > I believe the paper was just stating that the mixing distribution cannot be learned from data that represents a single draw of an elliptical process. However, the section seemed to imply that either mixture processes or non-stationary processes (or both?) cannot be estimated which would be helped by an illustrative example or two.
>
> You are correct that the main point we’re trying to make is that the mixing distribution is not identifiable from a single draw of an elliptical process. To clarify this and avoid confusion we (1) rephrased this section in terms of identifiability (2) added an example on learning the mixing distribution from multi-path data, see section 3.4.
>
> > What is the effect of this point of ergodicity on the experimental conclusions?
>
> The implication is that precise inference of an EP prior from single path data is impossible. Consequently, our experiments do not consider EP priors except on the new example with multi-path data. On the other hand, we consider EP posteriors in the regression experiments since the results of combining a GP prior with an elliptic likelihood is an elliptic posterior.
>
> > "The conditional distribution is guaranteed to be a consistent elliptical distribution but not necessarily the same as the original one..." How does this issue specifically manifest in inference?
>
> There are two ways to handle that the mixing distribution changes for the conditional EP. The first is to derive it analytically as shown in Appendix B, the second is to approximate $p(\xi|\mathbf{y})$ with a variational distribution. Since we, in practice, only use the second option in this paper, we moved the analytical derivation to the appendix.
>
> > Describe how the predictive mean (RHS of (16), (20)) is calculated for the experiments.
>
> To derive the predictive mean we use the same strategy as we do for a GP. We have added equation 17 to clarify this.

---

> > ### Comment · Reviewer_fi2A · 2022-07-25
> > **Better, but still lacking compelling supporting evidence**
> >
> > (First, my apologies to the authors for the very late response.)
> >
> > The experimental section is improved over the previous version of the paper:
> > 	* Two additional regression datasets are added with sizes > 20000.
> > 	* The competitor of Jokowiak et al. (2020) was added; a variant of the proposed method using an orthogonal component of the Jonkowiak paper was also added in the regression comparison.
> > 	* The overall readability of this section has been improved with clearer explanations of methods being compared and more standard figure formats.
> >
> > The updates help better position the proposed method in the landscape for large-scale probabilistic regression. I appreciate how the authors have addressed some of my previous concerns.
> >
> > I would really like to accept this paper, but the provided experimental results are still not compelling and some of the methodological choices are perplexing. When proposing a new method to solve a well-established problem, there should be clear evidence that shows the regimes in which the performance advantage of the proposed method is statistically significant. The clearest evidence for the proposed method is limited to synthetic settings. The real-world results are all either not significant, significant but suspect, or orthogonal to the stated advantage of the proposed method.
> >
> > Critical:
> >
> > * The reported results in Figure 7 for the competitor PP-GP-GP are different than those of the original paper Jonkowiak et al. (2020) wrt/ the matching datasets b/w the two papers. For NLL, elevators is slightly worse and kin40k and protein are significantly worse. These results (including the MSEs) need to be checked and differences to previously-reported results explained.
> >
> > * The classification experiment still only consists of small datasets (sample size < 1000). Sparse approximations are not required in this setting. If the paper wishes to only include small datasets, the proposed method and competitors should be evaluated as non-sparse methods. For comparison in truly non-sparse setting, there should be enough binary or multi-class classification datasets to choose from.
> >
> > * The illustrative examples for heavy-tailed homoscedastic noise and heteroscedastic noise (Figures 5, 6, and 9) are great, but why can't comparisons on real-world datasets be added here? Figure 11 seems to be along this line (but doesn't consider competitors).
> >
> > * The experiment labeled "multi-path" was interesting, but its inclusion appears motivated by a desire to discuss/address the EP prior identifiability issue raised in the paper. If the paper is proposing some version of a variational EP as a "drop-in" replacement for some version of a GP, then this experiment seems orthogonal and therefore out of scope. Note that adding this modeling capability as an additional contribution requires more discussion of related work (in this modeling aspect) and a proper experimental methodology that might include other approaches (for performing this kind of modeling).
> >
> > Non-critical:
> >
> > * "How does one select the normalizing flow parameterization for a novel dataset? We train the flow and model parameters together using stochastic gradient descent."
> >
> > I was referring to the number of bins used in the spline flow (described in Appendix G for the included experiments). How does one choose this setting for a novel dataset?
> >
> > In Appendix G: "The likelihood mixing distribution uses a spline flow with 9 bins and Softplus as its output transformation. The elliptic posterior mixing distribution uses a spline flow with 5 bins and a Sigmoid otput transformation."
> >
> > * In section 3.3, it is stated "To model this, we return to the idea of amortized variational inference and use a neural network with parameters \gamma_\omega to represent the mapping from input location to spline flow parameters..."
> >
> > and
> >
> > "...and learning a different spline flow with parameters dependent on both input location and noise..."
> >
> > The neural network and parameter-dependence of the spline flow utilized in the experiment need to be documented somewhere.

---

### Author Response · Authors · 2022-06-21
**Authors' initial response to the reviews**

We thank all three reviewers for their extraordinarily helpful and thorough reviews. We are encouraged to see that you found the overall idea interesting, and the paper well-written.

We have started working on the improvements and clarifications you've suggested, focusing on the motivation and experimental justification, and will respond to your reviews individually. We look forward to an engaging conversation.

---

### Author Response · Authors · 2022-07-03
**Revision uploaded**

We thank the reviewers again for their valuable comments which we believe have significantly improved the paper. Before individually responding to the points raised, we first give a high-level summary of the main changes in the revised version:

**Introduction**: extended the first paragraph to clarify the motivation and revised the contribution at the end.

**Background**: reworded the paragraph on ergodicity in terms of identifiability. Moved the nonessential parts of the paragraph on the conditional distribution to the appendix.

**Method**: did a major restructuring to (hopefully) make it clearer. Added details on the ELBO training, and also extended it with a derivation of a training method based on the work by Jankowiak et al. Added a section on applying the method to multi-path data.

**Experiments**:
- section 4.1: added figure 6 showing a comparison with a standard (exact) GP to illustrate why it can be misleading.
- section 4.2: included several larger datasets and comparisons of the two training methods as well as an exact GP (where computationally feasible). Changed the presentations of the results into figures instead of a table, and also included Table 1 to describe the different methods compared.
- section 4.3: changed the presentation of the results and included accuracy.
- section 4.5: a new section showing results using multi-path data.

**Related work** and **Conclusions**: minor changes.

---

### Author Response · Authors · 2022-07-28
**Plan for second revision**

We, again, thank the reviewers for their helpful and actionable comments. Our plan for the next revision is to:
- Remove the multi-path approach;
- Investigate the variational approximation further, in particular (i) how well it approximates the true posterior, and (ii) how it depends on the number of inducing points;
- Add a non-sparse competitor in the classification experiments;
- Add a real-world dataset with heavy-tailed homoscedastic noise or heteroscedastic noise;
- Retrain the GP baseline in a more standard way.

It is challenging to study the prior in isolation since we use the variational posterior for training. We are still contemplating ways around this entanglement, e.g. exploring priors with fixed mixing distributions of different widths.

In the regression experiments (Figure 7), the reason for the differences from Jankowiak et al. is that we’ve used a different number of inducing points and consistently used RBF kernels throughout the paper, whereas Jankowiak et al. used Matérn. We will clarify this.

---

> ### Comment · Reviewer_9XiL · 2022-07-30
> **Suggestion**
>
> I think there are two things you can do to answer the question "is the elliptical process really the thing that improves performance?"
>
> Firstly, you should compare to a Gaussian process that is as exact as possible. If you can, do a standard non-approximated GP. If you can't then Tisias's sparse variational method [1] method is great, if you add enough inducing points (you need to do a sweep over this parameter, and stop adding inducing points when the performance has stopped improving). Make sure that hyperparameters are also well-trained, i.e. by BFGS.
>
> Next, you can compare to your approximation method, but where the model you are approximating is a GP. This should give a very similar (or slightly worse, due to approximation error) result as the previous experiment. This tests whether your approximation method introduces any error that actually helps performance. This happens often! Most notably, in the FITC approximation, which can perform better than a normal GP because it can model heteroskedastic noise [2], even though this means it is a bad approximation to a normal GP [3]!
>
> Finally, you can compare to your approximation to an elliptical process. If this performs better than both, then this is compelling evidence that it really is the elliptical process change to the model that is helping.
>
> The strong GP baseline is the most important though. Unfortunately, it does take some effort to train a GP model properly. Somehow it seems more laborious than training deep learning models these days. However, all these steps to train a GP model properly are known and should be followed if one wants to argue for an improved model.
>
> [1] http://proceedings.mlr.press/v5/titsias09a/titsias09a.pdf
> [2] http://www.gatsby.ucl.ac.uk/~snelson/SPGP_up.pdf
> [3] https://arxiv.org/abs/1606.04820

---

### Decision · Action_Editors · 2022-08-12

**Recommendation:** Reject

**Comment:**

Overall, I concur with the reviewers that the paper is well-written and clear, and presents nice ideas that I believe will be of interest to many working on Gaussian processes. While I am recommending reject due to the need for some additional experiments (as detailed below), I believe that the changes needed should be fairly straightforward and hope to see a resubmission soon.

While the experiments have been notably improved since the first submission, they still do not succeed at addressing the fundamental question of whether using an elliptical process is beneficial over a Gaussian process (when stripped of the various approximation techniques used both in this paper, and in comparison methods), and what the impact of your approximations is. The comment from July 30 by Reviewer 9XiL lays out a proposed set of experiments to address this, which seem a manageable amount of additional work. I believe these results, if they do show a notable improvement over the GP, will really highlight the fundamental benefits of your approach, and will also highlight areas where future work could be beneficial.

I agree with your proposed changes and would like to see these in the resubmitted manuscript, particularly the proposed new experiments on real-world and larger datasets. While overall I agree with the decision to remove the multi-path results (as the reviewers say, it detracts from the main story; isn't really a standard use case of GPs; and is a little underexplored). I do think the discussions on identifiability are important though, and think it is fine to explain that identifiability can be achieved in a multi-path context.

I believe the paper is very close to acceptance, and I would hope to see a revised version accepted. The reason I am recommending reject rather than accept with minor revisions, is because of the need to properly evaluate the approximation and compare with a standard GP. If these experiments do not show a key difference in performance over the GP, then an accept with minor revisions would prove inappropriate. However, if those experiments show the desired results, I do not think that the changes needed to the paper are too dramatic.

---

> ### Author Response · Authors · 2022-08-24
> **Acknowledged**
>
> Thanks for the clear motivation and explanation for what is currently lacking. We understand your reasoning and are actively working on a resubmission. We hope to come back within a couple of weeks, but it depends on the time required to run the additional experiments.